# Time-independent Generalization Bounds for SGLD in Non-convex Settings

**Tyler Farghly**
Department of Statistics
University of Oxford
farghly@stats.ox.ac.uk

**Patrick Rebeschini**
Department of Statistics
University of Oxford
patrick.rebeschini@stats.ox.ac.uk

## Abstract

We establish generalization error bounds for stochastic gradient Langevin dynamics (SGLD) with constant learning rate under the assumptions of dissipativity and smoothness, a setting that has received increased attention in the sampling/optimization literature. Unlike existing bounds for SGLD in non-convex settings, ours are time-independent and decay to zero as the sample size increases. Using the framework of uniform stability, we establish time-independent bounds by exploiting the Wasserstein contraction property of the Langevin diffusion, which also allows us to circumvent the need to bound gradients using Lipschitz-like assumptions. Our analysis also supports variants of SGLD that use different discretization methods, incorporate Euclidean projections, or use non-isotropic noise.

## 1 Introduction

Investigating the generalization error of a learning algorithm is a fundamental problem in machine learning that has motivated the development of a rich theory connecting notions of model complexity and sensitivity to the ability of a learning algorithm to generalize well to unseen data. Recently, there has been increased interest in studying the generalization capabilities of stochastic gradient descent (SGD) and its variants, e.g. [13, 1, 6, 17, 22, 19]. Despite them being some of the most important methods of modern machine learning and statistics, their learning capabilities have not been fully explored, particularly in non-convex settings.

One variant that has attracted a lot of attention is stochastic gradient Langevin dynamics (SGLD) [29]. With the addition of independent Gaussian noise to each iteration, SGLD combines the SGD framework with the Langevin diffusion, a stochastic process that converges to the Gibbs distribution of a given objective function. By tuning the scale of the noise applied, SGLD has been shown to work well as both a sampling scheme and a statistical learning algorithm in a variety of settings, e.g. [29, 25, 4, 14]. Using its relationship with the Langevin equation, various tools from stochastic analysis have been adopted to develop a rich theory analyzing SGLD in settings where no comparable results seem to be available for SGD, e.g. [23, 32, 5, 31]. In particular, a recent work from Raginsky et al. [23] provides non-asymptotic excess risk bounds for SGLD for a broad class of non-convex learning problems and these bounds were derived using the exponential ergodicity of Langevin diffusions. Following this work, there has been ongoing progress in the development of bounds for both optimization error and mixing times, e.g. [32, 5, 31, 7, 4]. Many of these works are motivated by recent results showing that Langevin diffusions contract in particular Wasserstein distances [8, 9].

In parallel to this literature, there has been a growing body of work that considers generalization error bounds for SGLD. Using uniform stability, Mou et al. [17] showed that under the assumption of bounded gradient updates and bounded loss functions, SGLD attains generalization bounds that grow slower with time than those that are known for SGD in identical settings [13, 6]. By bounding

35th Conference on Neural Information Processing Systems (NeurIPS 2021).

the generalization error using the mutual information between its input and output [30], Pensia et al. [22] established similar results while replacing the assumption of bounded loss functions with the weaker assumption that the loss function applied to the data is subgaussian. This technique has also been used to develop data-dependent generalization bounds that depend on the gradient along the trajectory of the iterates [18, 12, 17, 19].

Although the assumptions made in the aforementioned bounds cover several settings of interest, we are not aware of any convergence analysis that considers Lipschitz, bounded, or subgaussian objective functions. In general, the Gibbs distribution, which is an object that is fundamental to the design and known convergence analyses for SGLD [29, 25, 23, 5, 32], cannot be defined in the case that the objective is bounded over the Euclidean space, as the distribution would not integrate to one. Similarly, we are not aware of any generalization bounds for SGLD that use the assumptions of dissipativity and smoothness that are consistently applied in the non-convex sampling/optimization literature, e.g. [23, 5, 32].

Another commonality in existing generalization bounds for SGLD is that they grow indefinitely with time. Sampling from the Gibbs distribution, the limiting behavior of the Langevin diffusion, is known to be uniformly stable in multiple settings [30, 23] and it has also been shown that the continuous-time Langevin dynamics are uniformly stable when applied to bounded Lipschitz functions with weight decay [16]. However, convergence in generalization error for SGLD seems to have only been observed in preliminary empirical evidence that uses data-dependent bounds [18, 12]. Time-independent bounds have been obtained for SGD, for example, under the assumption of strong convexity in [13] and in non-convex settings, Lei and Ying [15] obtain bounds that decrease with time but do not decay to zero as the sample size increases.

## 1.1 Our contributions

In this paper, we obtain expected generalization error bounds for SGLD for learning problems under dissipativity and smoothness assumptions. We analyze both the discrete-time algorithm as well as a continuous-time version and in both cases, we obtain bounds that converge with time. Taking the supremum with respect to time yields time-independent bounds that, with the appropriate scaling of the learning rate, decay to zero as the sample size increases.

At first, we focus on the special case of Lipschitz loss functions with weight decay, which is the primary example of a dissipative objective given in Raginsky et al. [23]. For the continuous-time algorithm, we obtain a bound with rate $O(n^{-1})$ and for the discrete-time algorithm, we obtain a bound with the slower rate $O(n^{-1} + \eta^{1/2})$, where $\eta$ is the learning rate. Then we extend the result to the full dissipative case without Lipschitz requirements. In this setting, we obtain bounds with rate $O(n^{-1}\eta^{-1/2})$ for the continuous-time algorithm and with rate $O(n^{-1}\eta^{-1/2} + \eta^{1/2})$ in the discrete-time case. We also discuss how our method allows for the consideration of different discretization techniques and how it can be used to obtain generalization error bounds for modifications of SGLD that incorporate Euclidean projections or non-isotropic noise. From our bounds, a scheme for choosing $\eta$ follows: in the Lipschitz setting, we obtain dimension-free $\mathcal{O}(n^{-1})$ bounds by setting $\eta \propto n^{-2}d^{-1}$ where $d$ is the model dimension, and in the dissipative setting we find that a scaling of $\eta \propto n^{-1}$ leads to $O(n^{-1/2})$ bounds that, in general, scale exponentially in dimension.

To derive time-independent bounds, our proof technique depends fundamentally on the two sources of noise that occur in the algorithm: the random mini-batches and the injected Gaussian noise. Using the framework of uniform stability, in the sense defined in [10], we derive generalization bounds by estimating how much SGLD diverges in Wasserstein distance when an element of the data set is changed. We exploit recent results that use reflection couplings to show that under dissipativity-type assumptions, Langevin diffusions contract in Wasserstein distance. Though this property has been adopted extensively in the sampling/optimization literature, we are not aware of any results for generalization bounds based on this property. Using the convexity of the Wasserstein distance, we combine this with the stability induced by using stochastic gradients to obtain bounds that are time-independent.

A peculiarity of our results that arises from the methodology we use is that some of our bounds diverge as $\eta \to 0$. This contrasts with the usual approach of stability-based generalization bounds based on non-expansivity that often require the learning rate to decay sufficiently fast or the time-horizon to be sufficiently small to guarantee bounds that are non-vacuous [13, 17]. Furthermore, for our bounds to

| Paper | Assumptions | EGE Bound | Key | |
|---|---|---|---|---|
| Raginsky et al. [23] | D, S | $\mathcal{O}(\eta t + e^{-\eta t/c} + 1/n)$ | B | Bounded |
| Mou et al. [17] | B, L | $\mathcal{O}((\eta t)^{1/2}/n)$ | L | Lipschitz |
| Mou et al. [17] | L, SG, $\ell_2$ | $\mathcal{O}(\log(t+1)/n)^{1/2}$ | S | Smooth |
| Pensia et al. [22] | L, SG | $\mathcal{O}(\eta t/n)^{1/2}$ | $\ell_2$ | Weight decay |
| **Present work** | L, S, $\ell_2$ | $\mathcal{O}((\eta t \wedge 1)(1/n + \eta^{1/2}))$ | D | Dissipative |
| | D, S | $\mathcal{O}((\eta t \wedge 1)(\eta^{-1/2}/n + \eta^{1/2}))$ | SG | Subgaussian |

Table 1: Comparison of expected generalization error bounds for SGLD in recent works.

converge with time, we require the mini-batch size to be less than $n$. While our bounds support the case of full-batch gradient descent, we find that our bounds grow indefinitely with time.

## 2   Background and notation

### 2.1   Stability and generalization

In this paper, we consider a loss function $f : \mathbb{R}^d \times \mathcal{Z} \to \mathbb{R}$ where the Euclidean space $\mathbb{R}^d$ represents the set of possible model parameters and $\mathcal{Z}$ represents the data instance space. A common objective in learning theory is to minimize the *population risk* which, given a data distribution $P$, is defined by

$$F_P(x) := \mathbb{E}_{Z \sim P} f(x, Z).$$

In practice, we often cannot compute $F_P(x)$ so we instead collect independent samples from $P$ to form a *data set* and we compute the average loss, or *empirical risk*, over the data set. We will use the notation $S = (z_1, ..., z_n)$ to denote the data set and define the empirical risk as

$$F_S(x) := \frac{1}{n} \sum_{i=1}^{n} f(x, z_i).$$

When the parameter is chosen by a random algorithm that depends on the data set, denoted $A(S)$, we define the object central to this paper, the *generalization error*, as follows:

$$\text{gen}(A) := F_P(A(S)) - F_S(A(S)).$$

To bound this quantity we employ the following notion of *uniform stability*.

**Definition 2.1** ([13], Definition 2.1). *An algorithm $A$ is $\varepsilon$-uniformly stable if*

$$\varepsilon_{stab}(A) := \sup_{S \cong \widehat{S}} \sup_{z \in \mathcal{Z}} \mathbb{E}\Big[ f\big(A(S), z\big) - f\big(A(\widehat{S}), z\big) \Big] \leq \varepsilon,$$

*where the first supremum is over data sets $S, \widehat{S} \in \mathcal{Z}^n$ that differ by one element, denoted by $S \cong \widehat{S}$.*

The connection between generalization and stability under changes in the data set has received increased attention since the paper of Bousquet and Elisseeff [2], and these results have been extended to account for random algorithms by Elisseeff et al. [10]. The precise notion of stability that we consider in this paper is given in Hardt et al. [13].

**Proposition 2.2** ([13], Theorem 2.2). *Suppose $A$ is an $\varepsilon$-uniformly stable algorithm, then the expected generalization error is bounded by*

$$\big| \mathbb{E}_{A,S} \, \text{gen}(A) \big| \leq \varepsilon.$$

Though we will present the results of this paper as bounds on the expected generalization error, uniform stability bounds can also give high probability bounds for the generalization error [11, 3]. Outside of generalization bounds, the concept of uniform stability has also been shown to be fundamentally related to differential privacy and learnability [28, 24].

## 2.2 Stochastic gradient Langevin dynamics

We will now define the algorithm of interest in this paper. Given a *mini-batch* $B \subset [n] := \{1, ..., n\}$, let the *mini-batch average* be defined by

$$F_S(x, B) := \frac{1}{|B|} \sum_{i \in B} f(x, z_i).$$

In our analysis we will consider uniformly sampled random mini-batches of fixed size $k \leq n$. We now define SGLD, which given an initial distribution $\mu_0$, is characterized by the update

$$x_{t+1} = x_t - \eta \nabla F_S(x_t, B_{t+1}) + \sqrt{2\beta^{-1}\eta}\xi_{t+1}, \quad x_0 \sim \mu_0, \tag{1}$$

where $(B_t)_{t=1}^\infty$ is an i.i.d. sequence of random variables distributed uniformly on the set $\{B \subset [n] : |B| = k\}$ and $(\xi_t)_{t=1}^\infty$ is an i.i.d. sequence of $N(0, I_d)$ standard Gaussian random variables. The parameters $\eta, \beta > 0$ are tunable and are referred to as the *learning rate* and *inverse temperature* respectively. Here and throughout the paper we use $\nabla$ to refer to the gradient with respect to the model parameter.

If $k = n$ then (1) describes the Euler-Maruyama discretization of the (overdamped) Langevin equation with potential $F_S(\cdot)$. This is a stochastic differential equation of the form

$$dX_t = -\nabla F_S(x)dt + \sqrt{2\beta^{-1}}dW_t, \quad X_0 \sim \mu_0,$$

where $W_t$ is a $d$-dimensional Wiener process [21]. In the more common case that $k < n$ we can still define a stochastic differential equation that approximates SGLD, given by

$$dX_t = -\nabla F_S(X_t, B_{\lceil t/\eta \rceil})dt + \sqrt{2\beta^{-1}}dW_t, \quad X_0 \sim \mu_0. \tag{2}$$

We will refer to the solution $X_t$ as the *continuous-time* SGLD algorithm. Note that, under the smoothness assumptions imposed throughout this paper, all stochastic differential equations considered have strong solutions (see Theorem 3.1 of [21]).

Since $x_t$ is a Markov process, we can define its Markov kernel $R_x$. Though we will not frequently reference this fact, we will use the notation $\mu R_x^s$ to denote the law of $x_{t+s}$ under the condition $x_t \sim \mu$. Because of its dependence on $B_k$, $X_t$ is not necessarily a Markov process in continuous-time, but the discrete-time process $(X_{t\eta})_{t=0}^\infty$ does in fact satisfy the Markov property and so we denote its kernel by $R_X$.

Given a deterministic set $B \subset [n]$ we use the notation $\mu P_t^B$ to denote the law of $\theta_t$, the solution to the SDE,

$$d\theta_t = -\nabla F_S(\theta_t, B)dt + \sqrt{2\beta^{-1}}dW_t, \quad \theta_0 \sim \mu, \tag{3}$$

which is a Markov (diffusion) process. Thus $\mu R_X$ can be computed by integrating $\mu P_\eta^B$ over $B$ with respect to the mini-batch distribution.

Since this work is concerned with stability under changes in the data set, we will also be interested in SGLD when $S$ is replaced with some data set $\widehat{S}$ that differs by a single element. We will use the notation $\widehat{x}_t$ and $\widehat{X}_t$ to denote the respective counterparts of $x_t$ of $X_t$ when trained using $\widehat{S}$ instead of $S$ and similarly we will use $\widehat{R}_x$, $\widehat{R}_X$ and $\widehat{P}_t^B$.

## 2.3 Wasserstein distance

For probability measures $\mu, \nu$ on $\mathbb{R}^d$ with finite $p^{th}$ moment, we define the *Wasserstein distance*

$$\mathrm{W}_p(\mu, \nu) := \left( \inf_{\pi \in \mathcal{C}(\mu,\nu)} \int \|x - y\|^p \, \pi(dx, dy) \right)^{1/p},$$

where $\|\cdot\|$ is the Euclidean norm. Here the infimum is over all couplings of $\mu$ and $\nu$, that is, the set of all probability measures with marginals $\mu$ and $\nu$. In our analysis, we will also consider Wasserstein distances based on any semimetric $\rho$, which under certain integrability assumptions is defined by

$$\mathrm{W}_\rho(\mu, \nu) := \inf_{\pi \in \mathcal{C}(\mu,\nu)} \int \rho(x, y) \, \pi(dx, dy).$$

Note that throughout this paper all probability measures considered outside of the data distribution, are Borel probability measures on $\mathbb{R}^d$.

A property of the Wasserstein distance that will prove to be of central importance in our results is its convexity.

**Lemma 2.3** (Convexity of the Wasserstein distance). *Suppose that $\rho$ is a semimetric and $\mu_1, \mu_2, \nu_1, \nu_2$ are probability measures. Then, for any $r \in [0, 1]$,*
$$W_\rho(\mu, \nu) \le r\, W_\rho(\mu_1, \nu_1) + (1 - r)\, W_\rho(\mu_2, \nu_2),$$
*where we define $\mu(dx) = r\mu_1(dx) + (1-r)\mu_2(dx)$ and $\nu(dx) = r\nu_1(dx) + (1-r)\nu_2(dx)$.*

The proof of this result, as well as a more general statement, can be found in [26], Theorem 4.8.

## 3 The Lipschitz setting with weight decay regularization

To begin our analysis, we focus on a special case of the dissipativity and smoothness assumptions. We will assume that the loss function $f(x, z)$ is Lipschitz in $x$ for any fixed $z$, and when performing gradient updates we will use the empirical risk with an added weight decay term:
$$\widetilde{F}_S(x, B) = F_S(x, B) + \frac{\lambda}{2}\|x\|^2.$$
This setting has been considered in the data-dependent literature for SGLD, for example by Mou et al. [17], and under the additional assumption of boundedness, this setting has also been considered in the stability-based analysis of the continuous Langevin dynamics by Li et al. [16].

The following assumptions are imposed:

**Assumption 3.1.** *For each $z \in \mathcal{Z}$, $f(\cdot, z)$ is $L$-Lipschitz: for all $x_1, x_2 \in \mathbb{R}^d$ and $z \in \mathcal{Z}$,*
$$|f(x_1, z) - f(x_2, z)| \le L\|x_1 - x_2\|.$$

**Assumption 3.2.** *For each $z \in \mathcal{Z}$, $f(\cdot, z)$ is differentiable and $M$-smooth: for all $x_1, x_2 \in \mathbb{R}^d$ and $z \in \mathcal{Z}$,*
$$\|\nabla f(x_1, z) - \nabla f(x_2, z)\| \le M\|x_1 - x_2\|.$$

**Assumption 3.3.** *The initial condition $\mu_0$ has finite first moment $\sigma_1 := \mu_0(\|\cdot\|) < \infty$.*

When using weight decay, Assumption 3.1 guarantees that $\widetilde{F}_S$ is dissipative which allows us to obtain Wasserstein contractions. Assumption 3.2 guarantees the existence of a solution to the SDE in (2) and allows us to control the error that is brought about when using it to approximate the discrete-time algorithm. Assumption 3.3 is used to guarantee that SGLD has finite first moment so that we can perform our analysis using the 1-Wasserstein distance. Note that the weight decay term only affects the gradient updates of SGLD and is not used in the computation of the generalization error, which is still defined in terms of $F_S$ and $F_P$ and not $\widetilde{F}_S$.

**Theorem 3.1.** *Suppose Assumptions 3.1–3.3 hold and $\eta < 1$, then for any $t \in \mathbb{N}$ the continuous-time algorithm attains the generalization bound*
$$|\mathbb{E}\operatorname{gen}(X_{\eta t})| < C_2 \min\left\{\eta t, \frac{(C_1 + 1)n}{n - k}\right\}\frac{k}{n}.$$
*Furthermore, if $\eta < \lambda^{-1}$ then the discrete-time algorithm attains the generalization bound*
$$|\mathbb{E}\operatorname{gen}(x_t)| < C_3 \min\left\{\eta t, \frac{(C_1 + 1)n}{n - k}\right\}\left(\frac{k}{n} + (\eta d)^{1/2}\right).$$
*The positive constants $C_1 \equiv C_1(M, L, \lambda, \beta), C_2 \equiv C_2(M, L, \lambda, \beta), C_3 \equiv C_3(M, L, \lambda, \beta, \sigma_1)$ are given in equations (5), (7) and (8) respectively.*

**Remark 3.2.** *This result suggests that by using the step-size scaling $\eta = \mathcal{O}(1/n^2 d)$ we obtain dimension-free time-independent generalization bounds for SGLD that scale with $\mathcal{O}(k/n)$.*

The bound for the continuous-time algorithm has no explicit dependence on dimension and in the discrete-time case the bound grows with the rate $d^{1/2}$. In general, the constants $C_1, C_2$ and $C_3$ depend exponentially on $M, L, \lambda$ and $\beta$. However, in the case of $\lambda \ge M$, the constants become polynomial in these parameters and $C_2$ becomes independent of $\beta$. A similar dependence was found in the time-independent bounds for the Langevin equation [16] and it is also found in convergence analyses in non-convex settings, leading to the time horizon required to attain a certain guaranteed accuracy having to scale exponentially with parameters including $d$ and $\beta$, e.g. [23, 5, 32].

## 3.1 Methodology

In this section, we present the method used to obtain the bounds in Theorem 3.1, in particular we proceed with the stability framework presented in Section 2.1. Without loss of generality, we require that the two data sets $S$ and $\widehat{S}$ differ only in the $n^{th}$ coordinate.

Under Assumption 3.1, to obtain uniform stability, it is sufficient to control the quantities $W_1(\mu_0 R_X^t, \mu_0 \widehat{R}_X^t)$ and $W_1(\mu_0 R_x^t, \mu_0 \widehat{R}_x^t)$. It follows directly from the definition of $\varepsilon_{stab}$ that

$$\varepsilon_{stab}(A) \leq L \sup_{S \cong \widehat{S}} W_1 \left( law\big(A(S)\big), law\big(A(\widehat{S})\big) \right). \tag{4}$$

A property similar to this has been utilized in [23], where it was used to prove the stability of the Gibbs algorithm.

To prove Theorem 3.1, we begin with a weak estimate for the divergence in 1-Wasserstein distance using synchronous couplings between $X_t$ and $\widehat{X}_t$, that is, we couple the processes by having them share the same Brownian motion.

**Lemma 3.3.** *Suppose Assumption 3.1 holds, then for any two probability measures $\mu, \nu$ on $\mathbb{R}^d$,*

$$W_1\left(\mu P_t^B, \nu \widehat{P}_t^B\right) \leq W_1(\mu, \nu) + 2Lt.$$

Additionally, via a similar technique, we obtain discretization error bounds.

**Lemma 3.4.** *Suppose Assumptions 3.1-3.3 hold and $\eta < \lambda^{-1}$, then with $\mu = \mu_0 R_x^t$ for any $t \in \mathbb{N}$,*

$$W_1(\mu R_x, \mu R_X) \leq \eta(\lambda + M)\left[\eta(\lambda\sigma_1 + 2L) + 2\sqrt{2d\beta^{-1}\eta}\right]\exp(M+1).$$

We refer to Appendix B for the proofs of these two lemmas.

In the case $\widehat{P}^B = P^B$, we can obtain sharper bounds than those given in Lemma 3.3. In a recent result from Eberle [8], reflection couplings were used to show that under conditions not dissimilar from dissipativity, diffusion processes contract in Wasserstein distance. The Wasserstein distance considered in this result does not use a standard metric, it uses the metric $\rho_g(x, y) := g(\|x - y\|)$ where $g$ is a strictly-increasing concave function that is constructed depending on the objective function (or drift term) used. In the lemma that follows we give a special case of such a result in the setting that we consider. Throughout this section we use the notation $W_g(\mu, \nu) := W_{\rho_g}(\mu, \nu)$. We refer to Appendix C for the proof and a broader discussion on this result.

**Lemma 3.5.** *Suppose Assumptions 3.1 and 3.2 hold, then there exists a strictly-increasing concave function $g : \mathbb{R}^+ \cup \{0\} \to \mathbb{R}^+ \cup \{0\}$ such that for any two probability measures $\mu, \nu$ on $\mathbb{R}^d$, any $B \subset [n]$ and $t \geq 0$, we have*

$$W_g\left(\mu P_t^B, \nu P_t^B\right) \leq e^{-t/C_1} W_g(\mu, \nu), \quad C_1 := \begin{cases} c_1 c_2, & \text{if } \lambda < M, \\ c_3, & \text{if } \lambda \geq M, \end{cases} \tag{5}$$

*with constants $c_1 = \exp[2\beta L^2(M - \lambda)/\lambda^2]$, $c_2 = 8(L^2\beta/\lambda + 1)/\lambda$, $c_3 = \max(16L^2\beta/\lambda^2, 2/\lambda)$. Furthermore, $W_g$ is equivalent to the $1$-Wasserstein distance in the sense that*

$$\frac{1}{2\max(c_1, 1)} W_1(\mu, \nu) \leq W_g(\mu, \nu) \leq W_1(\mu, \nu).$$

In the case that $n$ is not in the random mini-batch, i.e. $n \notin B$, which occurs with probability $1 - k/n$, it follows that $\widehat{P}^B = P^B$ and so the processes $X_{t\eta}$ and $\widehat{X}_{t\eta}$ contract with respect to the Wasserstein distance $W_g$. In the case that $n$ is in the random mini-batch, $n \in B$, Lemma 3.3 gives uniform bounds for how much $X_{t\eta}$ and $\widehat{X}_{t\eta}$ can diverge. The remainder of the proof works to combine these facts and shows that if $n$ is sufficiently large, $X_{t\eta}$ and $\widehat{X}_{t\eta}$ can only diverge by some fixed amount. We combine the two cases using the convexity of the Wasserstein distance and we find that given a sufficiently large $n$, the resulting bound does indeed converge over time.

*Proof of Theorem 3.1.* As was pointed out in (4), it is sufficient to bound the 1-Wasserstein distance between the processes to obtain stability bounds, so this is how we will proceed. Since the probability

of a random mini-batch containing the element $n$ is $k/n$, Lemma 2.3 is used to bound the Wasserstein distance with the decomposition

$$W_g\left(\mu R_X, \nu \widehat{R}_X\right) \leq \frac{k}{n} \sup_{B:n \in B} W_g\left(\mu P_\eta^B, \nu \widehat{P}_\eta^B\right) + \left(1 - \frac{k}{n}\right) \sup_{B:n \notin B} W_g\left(\mu P_\eta^B, \nu \widehat{P}_\eta^B\right).$$

Bounding the first term using Lemma 3.3 and the second term using Lemma 3.5, it follows that $W_g(\mu R_X, \nu \widehat{R}_X) \leq \tilde{c}_1 W_g(\mu, \nu) + \tilde{c}_2$ where $\tilde{c}_1 := \frac{k}{n} + (1 - \frac{k}{n})e^{-\eta/C_1}$ and $\tilde{c}_2 = 2L\eta\frac{k}{n}$. It follows by induction that

$$W_g\left(\mu_0 R_X^t, \mu_0 \widehat{R}_X^t\right) \leq \sum_{s=1}^t \tilde{c}_1^{t-s}\tilde{c}_2 + \tilde{c}_1^t W_g(\mu_0, \mu_0) = \frac{1 - \tilde{c}_1^t}{1 - \tilde{c}_1}\tilde{c}_2, \tag{6}$$

since $\tilde{c}_1 < 1$. Combining this with (4), it follows that

$$\varepsilon_{stab}(X_{\eta t}) \leq 2L(c_1 \vee 1) W_g\left(\mu R_X^t, \nu \widehat{R}_X^t\right) \leq C_2\eta\left(\frac{1 - \tilde{c}_1^t}{1 - \tilde{c}_1}\right)\frac{k}{n},$$

where,
$$C_2(M, L, \lambda, \beta) := 4L^2(c_1 \wedge 1). \tag{7}$$

This bound is simplified with the approximation $1 - \tilde{c}_1^t \leq 1 \wedge (1 - \tilde{c}_1)t$ and, using the bound $e^x \geq 1 + x$, it follows that $(1 - e^{-x})^{-1} \leq 1 + 1/x$ for $x > 0$ and so

$$\frac{1}{1 - \tilde{c}_1} = \frac{1}{(1 - k/n)(1 - e^{-\eta/c})} \leq \frac{n(c/\eta + 1)}{n - k}.$$

This result is extended to the discrete-time case by applying the triangle inequality,

$$W_g\left(\mu R_x, \nu \widehat{R}_x\right) = W_g\left(\mu R_x, \mu R_X\right) + W_g\left(\mu R_X, \nu \widehat{R}_X\right) + W_g\left(\nu \widehat{R}_X, \nu \widehat{R}_x\right).$$

The first and third terms are bounded using the discretization error bound in Lemma 3.4 and the same argument is applied to obtain the bound,

$$\varepsilon_{stab}(x_t) \leq 2L(c_1 \wedge 1) W_g\left(\mu_0 R_x^t, \mu_0 \widehat{R}_x^t\right) \leq C_3\eta\frac{1 - \tilde{c}_1^t}{1 - \tilde{c}_1}\left(\frac{k}{n} + (d\eta)^{1/2}\right).$$

where
$$C_3(M, L, \lambda, \beta, \sigma_1) := 4L(c_1 \vee 1)\left(L + (\lambda + M)\left(\lambda\sigma_1 + 2L + 2\sqrt{2\beta^{-1}}\right)\right). \tag{8}$$

$\square$

**Remark 3.6.** *For the discrete-time algorithm, the bound is obtained by appending the one-step discretization error to the bound in (6) for the continuous-time algorithm. Thus, the result can easily be extended to a broader range of discretizations of $(X_t)_{t \geq 0}$. Suppose $K_x$ is a Markov kernel on $\mathbb{R}^d$ such that for any probability measure $\mu$ on $\mathbb{R}^d$, $W_1(\mu K_x, \mu R_X) \leq \delta(\eta)$, then we immediately obtain the bound*

$$W_1\left(\mu_0 K_x^t, \mu_0 \widehat{K}_x^t\right) \leq \mathcal{O}\left(\frac{1}{n} + \frac{\delta(\eta)}{\eta}\right).$$

*In the case of the standard discrete-time algorithm we obtain $\delta(\eta) = O(\eta^{3/2})$. A simple example of a more accurate discretization follows by instead of approximating $X_\eta$ with a single Euler-Maruyama step we use $T = \lfloor 1/\eta \rfloor$ steps with step-size $\gamma = \eta/T$, then we can show that $\delta(\eta) \leq O(\sqrt{\gamma}) \leq O(\eta^2)$.*

**Remark 3.7.** *With our methodology, it is also possible to derive bounds for projected versions of SGLD. If $\Omega \subset \mathbb{R}^d$ is a compact set, we can define the algorithm,*

$$x_{t+1}^\Pi = \Pi_\Omega\left[x_t^\Pi - \eta\nabla F_S(x_t, B_{t+1}) + \sqrt{2\beta^{-1}\eta}\zeta_{t+1}\right], \tag{9}$$

*where $\Pi_\Omega$ is the Euclidean projection onto the set $\Omega$. A well-known property that is often used in the optimization literature is that for any $x_1, x_2 \in \mathbb{R}^d$, $\|\Pi_\Omega(x_1) - \Pi_\Omega(x_2)\| \leq \|x_1 - x_2\|$ and, given two measures $\mu, \nu$ on $\mathbb{R}^d$, this naturally extends to*

$$W_1(\mu^*\Pi_\Omega, \nu^*\Pi_\Omega) \leq W_1(\mu, \nu),$$

*where $\mu^*\Pi_\Omega$ denotes the push-forward of $\mu$ with respect to the mapping $\Pi_\Omega$. If we let $R_x^\Pi$ be the kernel of the process given in (9), it follows immediately that,*

$$W_1\left(\mu R_x^\Pi, \nu \widehat{R}_x^\Pi\right) \leq W_1\left(\mu R_x, \nu \widehat{R}_x\right),$$

*and hence the Wasserstein distance bound given in the proof also applies to the projected algorithm and therefore, the generalization bound in Theorem 4.1 also holds for the projected algorithm.*

**Remark 3.8.** *Our technique supports the case where, instead of assuming $\xi_t$ is a standard Gaussian random vector, we set $\xi_t \sim N(0, \Sigma)$ for some symmetric positive semidefinite matrix $\Sigma \in \mathbb{R}^{d \times d}$. If the operator norm of $\Sigma$ is 1, the conclusions of Lemma 3.3 and Lemma 3.5 still hold true and, as a result, the conclusion of Theorem 3.1 would also hold.*

## 4   The dissipative setting

Now we extend the results of the previous section to the full dissipative smooth case. We refer to Section 4 of [23] for a detailed discussion on dissipativity.

**Assumption 4.1.** *For each $z \in Z$, $f(\cdot, z)$ is $(m, b)$-dissipative: for all $x \in \mathbb{R}^d$ and $z \in \mathcal{Z}$,*

$$\langle \nabla f(x, z), x \rangle \geq m \|x\|^2 - b.$$

**Assumption 4.2.** *Same as Assumption 3.2.*

**Assumption 4.3.** *The initial condition $\mu_0$ has finite fourth moment $\sigma_4 := \mu_0(\|\cdot\|) < \infty$.*

Without the Lipschitz assumption, the analysis is more challenging as it is no longer sufficient to control the uniform argument stability or 1-Wasserstein distance. Furthermore, we can no longer guarantee that the difference between $F_S$ and $F_{\widehat{S}}$ is upper bounded.

**Theorem 4.1.** *Suppose Assumptions 4.1–4.3 hold. If $\eta \in (0, 1)$ then for any $t \in \mathbb{N}$, the continuous-time algorithm attains the generalization bound*

$$|\mathbb{E} \operatorname{gen}(X_{\eta t})| < C_5 \min\left\{\eta t, \frac{(C_4 + 1)n}{n - k}\right\} \frac{k}{n\eta^{1/2}}.$$

*Furthermore, if $\eta \leq 1/2m$, then the discrete-time algorithm attains the generalization bound*

$$|\mathbb{E} \operatorname{gen}(x_t)| < C_6 \min\left\{\eta t, \frac{(C_4 + 1)n}{n - k}\right\}\left(\frac{k}{n\eta^{1/2}} + \eta^{1/2}\right).$$

*The constants $C_4 \equiv C_4(M, m, b, d, \beta), C_5 \equiv C_5(M, m, b, d, \beta, \sigma_4), C_6 \equiv C_6(M, m, b, d, \beta, \sigma_4)$ are given in equations* (11), (12) *and* (13).

**Remark 4.2.** *Choosing $\eta = \mathcal{O}(n^{-1/2})$ leads to $\mathcal{O}(n^{-1/2})$ generalization bounds.*

In general, the constants $C_4, C_5$, and $C_6$ depend exponentially on parameters $M, m, b, d, \beta$. Improving the dependence on these parameters in specific settings would require improvements on estimates of the contraction rate and therefore, improvements on the coupling arguments given in [9].

A peculiarity of this result is that the bounds explode as $\eta \to 0$. This is due to the fact that when $\eta$ is smaller, there is less time for the processes to contract at each iteration. In the setting of Section 3, this is combated by the fact that the divergence bounds scale with rate $\mathcal{O}(\eta)$, but in the present setting we only obtain divergence bounds with rate $\mathcal{O}(\eta^{1/2})$. This is a result of the metric that we use in our analysis that is designed to suit the coupling arguments given in [9].

### 4.1   Methodology

Since this result relies on similar techniques to the proof of Theorem 4.1, we will postpone the proof to the appendix and in this section, we will focus on the challenges faced when extending to the dissipative setting. Without the Lipschitz assumption it is no longer sufficient to control the 1-Wasserstein distance and so we must turn to different metrics. Raginsky et al. [23] used continuity with respect to the 2-Wasserstein metric to derive stability bounds for the Gibbs sampler but, as has been noted in [8] and [27], contractions in 2-Wasserstein distance are notably more difficult to obtain than in 1-Wasserstein distance.

We use a semimetric of the form,

$$\rho(x, y) = g(\|x - y\|)(1 + 2\varepsilon + \varepsilon \|x\|^2 + \varepsilon \|y\|^2),$$

where $g : \mathbb{R}^+ \cup \{0\} \to \mathbb{R}^+ \cup \{0\}$ is a non-decreasing concave function and $\varepsilon \in (0, 1)$. As will be exhibited in the appendix, Eberle et al. [9] show that there exists $g$ such that for $\varepsilon$ sufficiently small,

$$\mathrm{W}_\rho\left(\mu P_t^B, \nu P_t^B\right) \leq e^{-t/C_4} \mathrm{W}_\rho\left(\mu, \nu\right),$$

for any two probability measures $\mu, \nu$ on $\mathbb{R}^d$. Additionally, this function is constant for $r > R$ and has the property $\varphi r \leq g(r) \leq r$ for some $\varphi, R \in \mathbb{R}^+$.

We show that this result can be used to derive stability bounds in the following lemma.

**Lemma 4.3.** *Suppose Assumptions 4.1 and 4.2 hold and let $A$ be a random algorithm, then*

$$\varepsilon_{stab}(A) \leq \frac{M(b/m + 1)}{\varphi \varepsilon (R \vee 1)} \sup_{S \cong \widehat{S}} \mathrm{W}_\rho \Big( law\big(A(S)\big), law\big(A(\widehat{S})\big)\Big).$$

The proof of Theorem 4.1 proceeds by establishing similar results to Lemma 3.3 and Lemma 3.5 using synchronous and reflection couplings. However, the argument is made markedly more difficult due to the fact that $\mathrm{W}_\rho$ is not a metric, we do not have access to the triangle inequality and it is not possible to show $\mathrm{W}_\rho$ is directly comparable to any standard Wasserstein distances.

## 5 Conclusion

In this paper, we derive time-independent generalization error bounds for SGLD under the assumptions of dissipativity and smoothness. We obtain bounds scaling as $O(n^{-1/2}\eta^{-1/2})$ for the continuous-time algorithm and scaling as $O(n^{-1/2}\eta^{-1/2} + \eta^{1/2})$ in the discrete-time case. In the special case of Lipschitz loss functions with weight decay regularization, we obtain faster rates of $O(n^{-1})$ for the continuous-time algorithm and $O(n^{-1} + \eta^{1/2})$ for the discrete-time case. In the latter case, we show that by having the step-size scaling $\eta = \mathcal{O}(d^{-1}n^{-2})$, we obtain dimension-free $\mathcal{O}(n^{-1})$-generalization bounds.

Within the framework of uniform stability, we use a combination of synchronous and reflection couplings to control the Wasserstein distance between versions of the algorithm with perturbations on the data set. Using the convexity of the Wasserstein distance, we show that the two versions of the algorithm can only diverge by a fixed amount that, with the appropriate scaling of $\eta$, decays to zero as $n$ increases.

The methodology used in our analysis allows for the extension to a broader class of discretizations. We are also able to explore modifications of the algorithm that incorporate Euclidean projections or non-isotropic Gaussian noise.

The fact that our proof relies heavily on properties of the Langevin diffusion does introduce some shortcomings. For example, the discretization error introduced when considering the discrete-time algorithm leads to an explicit dependence on the dimension and, furthermore, the dependence on model parameters aside from $d$ is often exponential. In addition, Gaussian noise is fundamental in the construction and analysis of the Langevin diffusion, so our analysis does not directly extend to algorithms where different types of noise are used.

## Acknowledgements

Tyler Farghly was supported by the Engineering and Physical Sciences Research Council (EP/T517811/1). Patrick Rebeschini was supported in part by the Alan Turing Institute under the EPSRC grant EP/N510129/1.

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
