# A  Properties of dissipative functions

In this section, we will briefly discuss some properties of the dissipativity assumption. Since the paper by Raginsky et al. [23], this assumption has seen frequent use in convergence analyses of SGLD in non-convex settings [5, 32, 31]. The primary motive for this assumption is that it guarantees that the Langevin equation has bounded moments – we recall this fact in the following lemma.

**Lemma A.1.** *Suppose Assumptions 4.1 and 4.2 hold and $\mu$ is a probability measure on $\mathbb{R}^d$, then for any $B \subset [n]$,*

$$\mu P_t^B(\|\cdot\|^p) \leq \mu(\|\cdot\|^p)e^{-pmt/2} + \left[2b/m + 2(p+d-2)/\beta m\right]^{p/2}(1 - e^{-pmt/2})$$

$$\leq \mu(\|\cdot\|^p) + \left[2b/m + 2(p+d-2)/\beta m\right]^{p/2}.$$

*Proof.* Suppose $\theta_t$ is a solution to the SDE in (3) with initial condition $\theta_0 \sim \mu$. From Itô's Lemma, it follows that

$$d\|\theta_t\|^p = -p\|\theta_t\|^{p-2}\langle\theta_t, \nabla F_S(\theta_t, B)\rangle dt + \beta^{-1}p(p+d-2)\|\theta_t\|^{p-2}dt$$
$$+ \sqrt{2\beta^{-1}}p\|\theta_t\|^{p-2}\langle\theta_t, dW_t\rangle$$
$$\leq -pm\|\theta_t\|^p dt + p\left[b + \beta^{-1}(p+d-2)\right]\|\theta_t\|^{p-2}dt + \sqrt{2\beta^{-1}}p\|\theta_t\|^{p-2}\langle\theta_t, dW_t\rangle$$

where the inequality follows from Assumption 4.1. This can be bounded further by

$$d\|\theta_t\|^p \leq -\frac{pm}{2}\|\theta_t\|^p dt + p\left[b + \beta^{-1}(p+d-2)\right]^{p/2}(m/2)^{1-p/2}dt + \sqrt{2\beta^{-1}}p\|\theta_t\|^{p-2}\langle\theta_t, dW_t\rangle.$$

Furthermore, using the product rule,

$$d\left(e^{pmt/2}\|\theta_t\|^p\right) \leq e^{pmt/2}p\left[b + \beta^{-1}(p+d-2)\right]^{p/2}(m/2)^{1-p/2}dt + e^{pmt/2}\sqrt{2\beta^{-1}}p\|\theta_t\|^{p-2}\langle\theta_t, dW_t\rangle,$$

and thus, by taking expectations it follows that,

$$\mathbb{E}\|\theta_t\|^p \leq \mathbb{E}\|\theta_0\|^p e^{-pmt/2} + p\left[b + \beta^{-1}(p+d-2)\right]^{p/2}(m/2)^{1-p/2}\int_0^t e^{-pm(t-s)/2}ds$$

$$\leq \mathbb{E}\|\theta_0\|^p e^{-pmt/2} + p\left[b + \beta^{-1}(p+d-2)\right]^{p/2}(m/2)^{1-p/2}\frac{2}{pm}(1 - e^{-pmt/2}).$$

$\square$

Under these assumptions, we can consider strongly convex functions and also a wide range of loss functions with multiple local minima. However, it is fairly restrictive in that it requires all local minima to be close to the origin. We show this in the lemma that follows. Throughout this section, we use the notation

$$B(x,r) := \{y \in \mathbb{R}^d : \|x - y\| < r\},$$

for any $x \in \mathbb{R}^d$ and $r > 0$.

**Lemma A.2.** *Suppose Assumption 4.1 holds and for each $z \in \mathcal{Z}$, $f(\cdot, z)$ is differentiable. Then for each $z \in \mathcal{Z}$, all local minima of $f(\cdot, z)$ are contained in the ball $\overline{B}(\mathbf{0}, \sqrt{b/m})$.*

*Proof.* Let $u \in \mathbb{R}^d$ be an arbitrary vector such that $\|u\| = 1$. Then for all $t > 0$,

$$\frac{d}{dt}f(ut, z) = \langle u, \nabla f(ut, z)\rangle \geq mt - \frac{b}{t}, \tag{10}$$

from which it follows that for $t > \sqrt{b/m}$, $\nabla f(ut, z) \neq \mathbf{0}$. Since $u$ is an arbitrary unit vector, this extends to $\nabla f(x, z) \neq \mathbf{0}$ if $\|x\| > \sqrt{b/m}$. $\square$

With this we can bound the gradient of the function at the origin uniformly over the instance space:

**Lemma A.3.** *Suppose Assumptions 4.1 and 4.2 hold, then for any $z \in \mathcal{Z}$,*

$$\|\nabla f(0, z)\| \leq M\sqrt{b/m}.$$

*Proof.* Another property of $f$ that follows from (10) is that for $t > R := \sqrt{b/m} + 1/m$,

$$\frac{d}{dt} f(ut, z) \geq 1$$

for any unit vector $u$. As a result, the quantity

$$\inf_{\|x\|=r} f(x, z),$$

also grows with a rate of at least 1 with respect to $r$ if $r > R$. Setting $r := R + \sup_{\|x\| \leq R} f(x, z) + 1$, which by the extreme value theorem must be a finite quantity, it follows that $\sup_{\|x\| \leq R} f(x, z) < \inf_{\|x\| \geq r} f(x, z)$.

We proceed by using this fact to show that $f(\cdot, z)$ must have global minima in the ball $B(\mathbf{0}, \sqrt{b/m})$. Consider the restriction of $f(\cdot, z)$ to the closed ball $\overline{B(\mathbf{0}, r)}$ and let $x^* \in \overline{B(\mathbf{0}, r)}$ be some point that attains the minimum value of this restriction (which must exist by the extreme value theorem). By our last deduction, it must hold that $\|x^*\| < r$ and in fact, $x^*$ minimizes the full function $f(\cdot, z)$. Furthermore, it must hold that $\nabla f(x^*, z) = \mathbf{0}$ and so by Lemma A.2, $\|x^*\| \leq \sqrt{b/m}$.

Finally, we apply the smoothness assumption to approximate the gradient at the origin:

$$\begin{aligned}
\|\nabla f(0, z)\| &\leq \|\nabla f(0, z) - \nabla f(x^*, z)\| + \|\nabla f(x^*, z)\| \\
&\leq M\|x^*\| + 0 \\
&\leq M\sqrt{b/m}.
\end{aligned}$$

$\square$

# B   Wasserstein bounds and moment estimates

To upper bound the Wasserstein distance between two probability measures $\mu$ and $\nu$, it is sufficient to consider any coupling $\pi \in \mathcal{C}(\mu, \nu)$ and use the inequality,

$$\mathrm{W}_\rho(\mu, \nu) \leq \mathbb{E}_{(X,Y)\sim\pi} \rho(X, Y).$$

Thus, we can design couplings such that the right-hand side is easily estimated.

In this section, we consider a type of coupling that is useful for bounding the distance between two diffusion processes. The *synchronous coupling* is formed by having the two processes solve SDEs with the same Brownian motion.

## B.1   Moment estimates

First, we will derive and recall some helpful moment estimates.

**Lemma B.1.** *Suppose Assumptions 3.1-3.3 hold and $\eta < \lambda^{-1}$, then for each $t \in \mathbb{N}$,*

$$\mu R_x^t(\|\cdot\|) \leq \mu(\|\cdot\|) + \frac{L + \sqrt{2\beta^{-1}d\eta^{-1}}}{\lambda}.$$

*Proof.* Let $x_t$ be an SGLD process with $x_0 \sim \mu$. We apply the Lipschitz property to deduce

$$\begin{aligned}
\mathbb{E}\|x_{k+1}\| &\leq \mathbb{E}\|x_k - \eta\nabla\widetilde{F}_S(x_k, B_{k+1})\| + \sqrt{2\beta^{-1}\eta}\mathbb{E}\|\xi_{k+1}\| \\
&\leq (1 - \lambda\eta)\mathbb{E}\|x_k\| + \eta L + \sqrt{2\beta^{-1}d\eta}.
\end{aligned}$$

Since $1 - \lambda\eta \in (0, 1)$, via an inductive argument we deduce that,

$$\begin{aligned}
\mathbb{E}\|x_k\| &\leq (1 - \lambda\eta)^k \mathbb{E}\|x_0\| + \frac{\eta L + \sqrt{2\beta^{-1}d\eta}}{1 - (1 - \eta\lambda)} \\
&\leq \mathbb{E}\|x_0\| + \frac{L + \sqrt{2\beta^{-1}d\eta^{-1}}}{\lambda}.
\end{aligned}$$

$\square$

Under the assumptions put forward in Section 4, we can obtain higher order moment estimates. To this end, we refer to a result by Chau et al. [5]:

**Lemma B.2** ([5], Lemma 3.9). *Suppose Assumptions 4.1-4.3 hold and $\eta < \frac{1}{2m}$, then*

$$\mu R_x^t(\|\cdot\|^{2p}) \leq \mu(\|\cdot\|^{2p}) + \tilde{c}(p),$$

$$\tilde{c}(p) = \frac{1}{m}\left(\frac{6}{m}\right)^{p-1}\left(1 + \frac{2^{2p}p(2p-1)d}{m\beta}\right)\left[\left(2b + 8\frac{M^2}{m^2}b\right)^p + 1 + 2\left(\frac{d}{\beta}\right)^{p-1}(2p-1)^p\right].$$

Note that Chau et al. take the maximum value of $\eta$ to be $\min\{m/2M^2, 1/m\}$ while we take the smaller bound $1/2m$. The fact that this is smaller follows from the fact that $M \geq m$ must hold.

## B.2 Divergence bounds

In this section we estimate the Wasserstein distance between $\mu P_t^B$ and $\nu\widehat{P}_t^B$ for any mini-batch $B \subset [n]$ of size $k$. We will do so using the synchronous coupling $(\theta_t, \widehat{\theta}_t)$, the solution to the system of SDEs

$$d\theta_t = -\nabla F_S(\theta_t, B)dt + \sqrt{2\beta^{-1}}dW_t,$$
$$d\widehat{\theta}_t = -\nabla F_{\widehat{S}}(\widehat{\theta}_t, B)dt + \sqrt{2\beta^{-1}}dW_t,$$

where $W_t$ is a $d$-dimensional Wiener process and $(\theta_0, \widehat{\theta}_0)$ is some coupling of $(\mu, \nu)$.

We begin by considering the setting of Section 3. Recall that in this section we had SGLD perform updates with the regularized objectives $\widetilde{F}_S$ and $\widetilde{F}_{\widehat{S}}$.

*Proof of Lemma 3.3.* Let $\pi$ be the coupling of $(\mu, \nu)$ that is optimal in the $W_1$-sense (existence is guaranteed by Theorem 4.1 of [26]). Furthermore, let $(\theta_t, \widehat{\theta}_t)$ be the synchronous coupling with initial condition $(\theta_0, \widehat{\theta}_0) \sim \pi$. From this, follows the decomposition

$$\theta_t - \widehat{\theta}_t = \theta_0 - \widehat{\theta}_0 + \int_0^t \left(\nabla\widetilde{F}_{\widehat{S}}(\widehat{\theta}_s, B) - \nabla\widetilde{F}_S(\theta_s, B)\right)ds$$

Applying the norm to both sides and taking expectations yields

$$\mathbb{E}\|\theta_t - \widehat{\theta}_t\| \leq \mathbb{E}\|\theta_0 - \widehat{\theta}_0\| + \int_0^t \mathbb{E}\|\nabla\widetilde{F}_{\widehat{S}}(\widehat{\theta}_s, B) - \nabla\widetilde{F}_S(\theta_s, B)\|ds$$

$$\leq \mathbb{E}\|\theta_0 - \widehat{\theta}_0\| + 2Lt.$$

The bound in the statement follows once it is noted that $\mathbb{E}\|\theta_0 - \widehat{\theta}_0\| = W_1(\mu, \nu)$ and $W_1(\mu P_t^B, \nu\widehat{P}_t^B) \leq \mathbb{E}\|\theta_t - \widehat{\theta}_t\|$. □

In the dissipative setting of Section 4 we will need to compute how far the process diverges from the initial condition.

**Lemma B.3.** *Suppose Assumptions 4.1 and 4.2 hold, then*

$$\mathbb{E}\|\theta_t - \theta_0\|^2 \leq 4M^2\left(\mathbb{E}\|\theta_0\|^2 + \frac{3b + 2d/\beta}{m}\right)t^2 + 4d\beta^{-1}t.$$

*Proof.* Using Jensen's inequality, we obtain the following decomposition:

$$\mathbb{E}\|\theta_t - \theta_0\|^2 \leq 2t\int_0^t \mathbb{E}\|\nabla F_S(\theta_s, B)\|^2ds + 4\beta^{-1}\mathbb{E}\|W_t\|^2.$$

Using Lemma A.3, the first term is bounded by

$$\mathbb{E}\|\nabla F_S(\theta_s, B)\|^2 \leq 2\mathbb{E}\|\nabla F_S(\theta_s, B) - \nabla F_S(0, B)\|^2 + 2\mathbb{E}\|\nabla F_S(0, B)\|^2$$

$$\leq 2M^2\mathbb{E}\|\theta_s\|^2 + 2M^2\frac{b}{m}$$

$$\leq 2M^2\mathbb{E}\|\theta_0\|^2 + 2M^2\frac{3b + 2d/\beta}{m},$$

where the final inequality follows from Lemma A.1. With this, it follows that

$$\mathbb{E}\|\theta_t - \theta_0\|^2 \le 4M^2 \Big(\mathbb{E}\|\theta_0\|^2 + \frac{3b + 2d/\beta}{m}\Big)t^2 + 4d\beta^{-1}t.$$

$\square$

## B.3 Discretization error bounds

In this section, we use synchronous-type couplings to obtain discretization error bounds. In particular, we will bound the Wasserstein distance between $\mu R_x$ and $\mu R_X$ for an arbitrary probability measure $\mu$.

By the convexity of the Wasserstein distance (see Lemma 2.3),

$$W_\rho(\mu R_x, \mu R_X) = \binom{n}{k}^{-1} \sum_{B \subset [n], |B| = k} W_\rho(\mu R_x^B, \mu P_\eta^B),$$

where $\mu R_x^B$ is the distribution of one step of (discrete-time) SGLD with fixed mini-batch $B \subset [n]$. Thus we consider an arbitrary mini-batch $B$ of size $k$ and seek to obtain bounds on $W_\rho(\mu R_x^B, \mu P_\eta^B)$. We define the relevant coupling $(\tilde{x}_t, \theta_{\eta t})$ for $t \in [0, 1]$ as follows:

$$d\theta_t = -\nabla F_S(\theta_t, B)dt + \sqrt{2\beta^{-1}}dW_t,$$
$$\tilde{x}_t = \tilde{x}_0 - \nabla F_S(x_0, B)\eta t + \sqrt{2\beta^{-1}}W_{\eta t},$$

where $\theta_0 \sim \mu$ and $\tilde{x}_0 = \theta_0$.

Once again, we will start by considering the setting of Section 3.

*Proof of Lemma 3.4.* For the first part of the lemma, we consider the coupling $(\tilde{x}_t, \theta_{\eta t})$ constructed above (but for regularized objective $\widetilde{F}_S$). By Jensen's inequality it follows that,

$$\|\theta_{\eta t} - \tilde{x}_t\| \le \eta \int_0^t \|\nabla \widetilde{F}_S(\theta_{\eta s}, B) - \nabla \widetilde{F}_S(\tilde{x}_0, B)\|ds$$

$$\le \eta(\lambda + M) \int_0^t \|\theta_{\eta s} - \tilde{x}_s\|ds + \eta(\lambda + M) \int_0^t \|\tilde{x}_s - \tilde{x}_0\|ds.$$

After taking expectations, the final term can be bounded using,

$$\mathbb{E}\|\tilde{x}_s - \tilde{x}_0\| \le \eta s \mathbb{E}\|\nabla \widetilde{F}_S(\tilde{x}_0, B)\| + \sqrt{2\beta^{-1}}\mathbb{E}\|W_{\eta s}\|$$

$$\le \frac{\eta s}{N} \sum_{i=1}^N \mathbb{E}\|\nabla f(\tilde{x}_0, z_i)\| + \sqrt{2d\beta^{-1}\eta s}$$

$$\le \eta s(\lambda \mathbb{E}\|\tilde{x}_0\| + L) + \sqrt{2d\beta^{-1}\eta s}.$$

Thus, by Grönwall's inequality

$$\mathbb{E}\|\theta_\eta - \tilde{x}_1\| \le \eta(\lambda + M)\Big[\eta(\lambda \mathbb{E}\|\tilde{x}_0\| + L) + \sqrt{2d\beta^{-1}\eta}\Big]\exp((\lambda + M)\eta).$$

Since $\mu = \mu_0 R_x^t$ for some $t$, we apply Lemma B.1 to deduce

$$\mathbb{E}\|\theta_\eta - \tilde{x}_1\| \le \eta(\lambda + M)\Big[\eta(\lambda \mu_0(\|\cdot\|) + 2L) + 2\sqrt{2d\beta^{-1}\eta}\Big]\exp((\lambda + M)\eta).$$

$\square$

The analogous result for the setting of Section 4 is derived with a similar technique:

**Lemma B.4.** *Suppose Assumptions 4.1 and 4.2 hold. Then, for any probability measure $\mu$ on $\mathbb{R}^d$, we have*
$$W_2(\mu R_x, \mu R_X)^2 \le 8\eta^3 \exp(2\eta^2 M^2)M^2(M^2\mu(\|\cdot\|^2) + M^2 b/m + \beta^{-1}d).$$

*Proof.* We proceed similarly to the proof of Lemma 3.4. By Jensen's inequality, it follows that

$$\mathbb{E}\|\theta_{\eta t} - \tilde{x}_t\|^2 \leq \frac{\eta^2}{t} \int_0^t \mathbb{E}\|\nabla F_S(\theta_{\eta s}, B) - \nabla F_S(\tilde{x}_0, B)\|^2 ds$$

$$\leq 2\frac{\eta^2}{t} M^2 \int_0^t \mathbb{E}\|\theta_{\eta s} - \tilde{x}_s\|^2 ds + 2\frac{\eta^2}{t} M^2 \int_0^t \mathbb{E}\|\tilde{x}_s - \tilde{x}_0\|^2 ds.$$

The second term is bounded using the smoothness assumption and Lemma A.3:

$$\mathbb{E}\|\tilde{x}_s - \tilde{x}_0\|^2 \leq 2\eta^2 s^2 \mathbb{E}\|\nabla F_S(\tilde{x}_0, B)\|^2 + 4\beta^{-1} d\eta s$$

$$\leq 4\eta^2 s^2 M^2 (\mathbb{E}\|\tilde{x}_0\|^2 + b/m) + 4\beta^{-1} d\eta s.$$

Thus it follows from Grönwall's inequality that

$$\mathbb{E}\|\theta_\eta - \tilde{x}_1\|^2 \leq \exp(2\eta^2 M^2) 8\eta^3 M^2 (M^2 \mu(\|\cdot\|^2) + M^2 b/m + \beta^{-1} d).$$

$\square$

## C   Wasserstein contractions and reflection couplings

Given two initial distributions $\mu$ and $\nu$, define the reflection coupling $(X_t, Y_t)$ by,

$$dX_t = -\nabla F_S(X_t, B)dt + \sqrt{2\beta^{-1}}dW_t,$$

$$dY_t = \begin{cases} -\nabla F_S(Y_t, B)dt + \sqrt{2\beta^{-1}}\big(I_d - 2e_t e_t^T\big)dW_t & \text{, if } t < T, \\ dX_t & \text{, if } t \geq T, \end{cases}$$

where we define the stopping time $T = \inf\{t \geq 0 : X_t \neq Y_t\}$, $(X_0, Y_0) \sim \pi$ for some coupling $\pi \in \mathcal{C}(\mu, \nu)$ and we define

$$e_t := (X_t - Y_t)/\|X_t - Y_t\|.$$

By Lévy's characterization of Brownian motion, it follows that the Itô integral of $\big(I_d - 2e_t e_t^T\big)dW_t$ does give a Brownian motion process. As with the synchronous coupling, this coupling is designed for analyzing the quantity $Z_t := X_t - Y_t$. Indeed, for $t < T$,

$$dZ_t = -(\nabla F_S(X_t, B) - \nabla F_S(Y_t, B))dt + 2\sqrt{2\beta^{-1}}Z_t/\|Z_t\|dW_t.$$

Furthermore, if we set $r_t = \|Z_t\|$ then by Itô's lemma, for any $t < T$,

$$dr_t = -r_t^{-1} Z_t \cdot (\nabla F_S(X_t, B) - \nabla F_S(Y_t, B))dt + 2\sqrt{2\beta^{-1}}dW_t.$$

For completeness, we will briefly discuss the two contraction results used in this paper.

### C.1   Contractions in 1-Wasserstein distance

In this section, we will discuss the technique used to obtain the result in [8] that leads to Lemma 3.5. In the paper by Eberle [8], they obtain exponential contractions between $X_t$ and $Y_t$ with respect to $W_g := W_{\rho_g}$ where $\rho_g$ is a metric defined by $\rho_g(x, y) = g(\|x - y\|)$ and $g$ is a strictly-increasing concave function. The contraction is obtained only under the condition that $\lim_{r \to \infty} \kappa(r) > 0$ where we define,

$$\kappa(r) = \beta \inf \left\{ \frac{\langle \nabla F_S(x, B) - \nabla F_S(y, B), x - y \rangle}{|x - y|^2} : \|x - y\| = r \right\}.$$

In the proof, they proceed by using Itô's Lemma to compute, for an arbitrary $g : \mathbb{R}^+ \cup \{0\} \to \mathbb{R}^+ \cup \{0\}$,

$$dg(r_t) \leq \beta^{-1}(4g''(r_t) - r_t \kappa(r_t)g'(r_t))dt + \sqrt{2\beta^{-1}}dW_t,$$

for each $t < T$. To obtain a contraction, they define a function $g$ such that $4g''(r) - r\kappa(r)g'(r) \leq -\beta g(r)/c$ holds for some $c > 0$. Under suitable integrability conditions this leads to,

$$\mathbb{E}g(r_t) \leq \mathbb{E}g(r_0) - c^{-1} \int_0^t \mathbb{E}g(r_s)ds.$$

From Grönwall's inequality, it follows that

$$\mathbb{E}\rho_g(X_t, Y_t) = \mathbb{E}g(r_t) \le e^{-t/c}\mathbb{E}g(r_0) = e^{-t/c}\mathbb{E}\rho_g(X_0, Y_0).$$

Eberle shows that under the condition $\lim_{r\to\infty}\kappa(r) > 0$, such a $g$ can be obtained for $c = \beta R_1^2/4\varphi_{min}$ where,

$$R_0 := \inf\{r' \ge 0; \kappa(r) \ge 0, \forall r \ge r'\}, \quad R_1 := \inf\{r' \ge R_0; \kappa(r)r'(r' - R_0) \ge 8, \forall r \ge r'\},$$

$$\varphi_{min} := \exp\left(-\frac{1}{4}\int_0^{R_0} s\kappa^-(s)ds\right).$$

We will not include the definition of $g$ in this paper for the sake of brevity, but we recall an important property: $\varphi_{min}/2 \le g' \le 1$. This will allow us to compare the $W_g$ metric with the 1-Wasserstein distance.

To simplify the constants given above, we will consider the case of Section 3 where we assume the loss function is $L$-Lipschitz and SGLD is performed using a weight decay regularized objective function, denoted by $\widetilde{F}_S$.

*Proof of Lemma 3.5.* From the Lipschitz and smoothness assumptions we obtain the estimates,

$$\langle\nabla\widetilde{F}_S(x, B) - \nabla\widetilde{F}_S(y, B), x - y\rangle \ge \lambda\|x - y\|^2 - M\|x - y\|^2,$$
$$\langle\nabla\widetilde{F}_S(x, B) - \nabla\widetilde{F}_S(y, B), x - y\rangle \ge \lambda\|x - y\|^2 - 2L\|x - y\|.$$

Since $\lambda r^2 - 2Lr \ge \frac{\lambda}{2}r^2$ for $r \ge 4L/\lambda$ we obtain

$$\kappa(r) \ge \begin{cases} -a, & \text{if } r < R, \\ b, & \text{if } r \ge R, \end{cases}$$

where $a = \beta(M - \lambda)$, $b = \beta\lambda/2$, $R = 4L/\lambda$. Since $R \ge R_0$ and $\kappa^- \le a$, an estimate of $\varphi_{min}$ is given by

$$\varphi_{min} \ge \exp\left(-\frac{aR^2}{8}\right).$$

To bound $R_1$ from above we will estimate $\widetilde{R}_1$ which is given by

$$\widetilde{R}_1 = \inf\{r' \ge R; br(r - R) \ge 8, \forall r \ge r'\}.$$

Since by assumption $\kappa(r)\widetilde{R}_1(\widetilde{R}_1 - R) \ge 8$ for each $r \ge \widetilde{R}_1$ and further $(\widetilde{R}_1 - R_0) \ge (\widetilde{R}_1 - R)$, we deduce that $R_1 \le \widetilde{R}_1$. This quantity can be computed to give,

$$\widetilde{R}_1 = \frac{R}{2} + \sqrt{\frac{R^2}{4} + \frac{8}{b}} \le R + \frac{2\sqrt{2}}{\sqrt{b}}.$$

It then follows that the contraction in the statement holds with rate

$$\frac{\beta R_1^2}{4\varphi_{min}} \le \frac{R^2 + 8/b}{2\beta^{-1}\exp(-\frac{aR^2}{8})} = c_1 c_2.$$

This result can be sharpened in the convex case $\lambda \ge M$. In Remark 5 of [8] they show that in this case $R_0 = 0$ and hence $\varphi_{min} = 1$. Furthermore, $R_1 \le \max(R, \sqrt{8/b})$ which leads to,

$$\frac{\beta R_1^2}{4\varphi_{min}} \le c_3.$$

$\square$

## C.2   Contractions under dissipativity

As noted in Section 4.1, obtaining contractions in the full dissipative case is more difficult. In this case, we consider the semimetric

$$\rho(x, y) := g(\|x - y\|)(1 + \varepsilon V(x) + \varepsilon V(y)),$$

where $\varepsilon < 1$, $g : \mathbb{R}^+ \cup \{0\} \to \mathbb{R}^+ \cup \{0\}$ is some concave, bounded and non-decreasing function and we define

$$V(x) := 1 + \|x\|^2.$$

The contraction result that we adopt is from a paper by Eberle et al. [9]. This result has previously been adopted in the same setting that we consider [5, 32] so we refer to the paper by Chau et al. [5] for a more detailed recollection.

Define the following constants:

$$R := 2\sqrt{(\beta d + \beta m + b)\Big(\frac{1}{\beta m} + 1\Big) - 1}, \quad \varphi := \frac{1}{2}\exp\Big(-\frac{\beta M}{2}R^2 - 2R\Big),$$

$$\varepsilon := 1 \wedge \varphi/R^2(\beta b + \beta m + d), \quad C_4 := \frac{\beta}{2}\big(\min\{\beta m/2, 2(\beta b + \beta m + d)\varepsilon, 2\varphi/R^2\}\big)^{-1}. \quad (11)$$

**Lemma C.1.** *Suppose Assumptions 4.1 and 4.2 hold. Then there exists a function g, such that for each $t \geq 0$,*

$$\mathrm{W}_\rho(\mu P_t^B, \nu P_t^B) \leq e^{-t/C_4}\,\mathrm{W}_\rho(\mu, \nu).$$

*Furthermore, g is constant on $[R, +\infty)$ and $\varphi r \leq g(r) \leq r$.*

In the paper by Eberle et al. [9], only the case of $\beta = 2$ is considered and so we must change our processes to suit this setting. If we have $A_t$ satisfy (2) and set $X_t = A_{\beta t/2}$, then it follows from Theorem 8.5.1 of [20] that $X_t$ satisfies

$$dX_t = \frac{\beta}{2}\nabla F_S(X_t, B)dt + dW_t.$$

As in the previous section, we have $(X_t, Y_t)$ be the reflection coupling for the above equation where $(X_0, Y_0) \sim \pi$ and $\pi$ is the $\mathrm{W}_\rho$-optimal coupling of $\mu$ and $\nu$.

Eberle et al. proceed in a similar fashion to what was laid out in the previous section, but the process for choosing $g$ is slightly different. Using the product rule, we can compute the following SDE for $\rho(X_t, Y_t)$:

$$d\rho(X_t, Y_t) = d(g(r_t)G(X_t, Y_t)) = G(X_t, Y_t)dg(r_t) + g(r_t)dG(X_t, Y_t) + [g(r), G(X, Y)]_t,$$

where we recall $r_t = \|X_t - Y_t\|$ and the final term is the covariation of $g(r_t)$ and $G(X_t, Y_t)$. This term can be estimated as follows:

$$d[g(r), G(X_t, Y_t)]_t = 4g'(r_t)\varepsilon|X_t - Y_t| \leq 4g'(r_t)G(X_t, Y_t).$$

Furthermore, from the bound on the generator of $X_t$,

$$\mathcal{L}V(x) = -\beta\langle x, \nabla F_S(x, B)\rangle + d \leq -\beta m\|x\|^2 + \beta b + d,$$

Eberle et al. estimate $dG(X_t, Y_t)$ by

$$dG(X_t, Y_t) \leq \big(2(\beta b + \beta m + d)\varepsilon\mathbb{1}_{r_t < R_1} - \min\{\beta m/2, 2(\beta b + \beta m + d)\varepsilon\}G(X_t, Y_t)\mathbb{1}_{r_t \geq R_2}\big)dt + dM_t^1$$

where $M_t^1$ is a martingale and $R_1, R_2$ are positive constants such that $0 < R_1 < R_2 < R$. Returning back to the product rule, we obtain

$$d\rho(X_t, Y_t) = \ G(X_t, Y_t)dg(r_t) + \big(2(\beta b + \beta m + d)\varepsilon\mathbb{1}_{r_t < R_1}g(r_t)$$
$$- \min\{\beta m/2, 2(\beta b + \beta m + d)\varepsilon\}\rho(X_t, Y_t)\mathbb{1}_{r_t \geq R_2} + 4g'(r_t)G(X_t, Y_t)\big)dt + g(r_t)dM_t^1.$$

What remains is designing a function $g$ such that the right hand side is less than $-c\rho(X_t, Y_t)$ for some $c > 0$. First, with Itô's Lemma and the smoothness assumption, it follows that

$$dg(r_t) \leq 2\big(\beta Mg'(r_t)r_t + g''(r_t)\big)dt + 2g'(r_t)\langle e_t, dW_t\rangle.$$

Though we will not explicitly include their construction, we remark that Eberle et al. construct a function $g$ such that

$$2\big(\beta M g'(r)r_t + g''(r)\big) \leq -2\varphi/R^2 g(r)\mathbb{1}_{r_t < R_2} - 2\varphi/R^2 g(r)\mathbb{1}_{r_t < R_1} - 4g'(r_t).$$

From this we deduce that for $\varepsilon \leq \varphi/R^2(\beta b + \beta m + d)$,

$$
\begin{aligned}
d\rho(X_t, Y_t) &\leq \big(-2\varphi/R^2\mathbb{1}_{r_t < R_2} - \min\{\beta m/2, 2(\beta b + \beta m + d)\varepsilon\}\mathbb{1}_{r_t \geq R_2}\big)\rho(X_t, Y_t) + dM_t^2 \\
&\leq -\min\big\{\beta m/2, 2(\beta b + \beta m + d)\varepsilon, 2\varphi/R^2\big\}\rho(X_t, Y_t) + dM_t^2
\end{aligned}
$$

for some martingale $M_t^2$. Thus, via the same argument used in Section C.1, we obtain the contraction,

$$\mathbb{E}\rho(X_t, Y_t) \leq e^{-\beta t/2C_4}\mathbb{E}\rho(X_0, Y_0).$$

Once we note that $W_\rho(\mu P_t^B, \nu P_t^B) \leq \mathbb{E}\rho(X_{2t/\beta}, Y_{2t/\beta})$ and $\mathbb{E}\rho(X_0, Y_0) = W_\rho(\mu, \nu)$, the contraction estimate given in the lemma immediately follows.

# D  Proof of Theorem 4.1

## D.1  Properties of the semimetric

Before we proceed with the proof of the theorem, we require some basic properties of the semimetric $\rho$. Recall that $\rho$ is defined by

$$\rho(x, y) = g(\|x - y\|)(1 + 2\varepsilon + \varepsilon\|x\|^2 + \varepsilon\|y\|^2),$$

where $g$ is concave and $g(r)$ is constant for $r > R$. Furthermore, we recall that $\varphi r \leq g(r) \leq r$

First, we prove Lemma 4.3, the $W_\rho$-continuity for functions of quadratic growth. A similar result is given in [23] for the 2-Wasserstein distance.

*Proof of Lemma 4.3.* Let $x, y \in \mathbb{R}^d$, then using the smoothness assumption it follows that for any $z \in \mathcal{Z}$,

$$
\begin{aligned}
f(x, z) - f(y, z) &= \int_0^1 \langle x - y, \nabla f((1-t)y + tx)\rangle dt \\
&\leq \int_0^1 \|x - y\|\Big(M\|(1-t)y + tx\| + \|\nabla f(0, z)\|\Big)dt \\
&= M\|x - y\|\Big(\sqrt{b/m} + \frac{\|x\|}{2} + \frac{\|y\|}{2}\Big),
\end{aligned}
$$

where, for the inequality, we have used Lemma A.3. Next, we use basic properties of the semimetric to show that this quantity is controlled by $\rho(x, y)$. For $\|x - y\| \leq R$, we use $f(\|x - y\|) \geq \varphi\|x - y\|$ to deduce

$$f(x, z) - f(y, z) \leq \frac{M}{\varphi}g(\|x - y\|)\Big(1 + \sqrt{b/m} + \frac{\|x\|^2}{2} + \frac{\|y\|^2}{2}\Big) \leq \frac{M(\sqrt{b/m} \wedge 1)}{2\varphi\varepsilon}\rho(x, y)$$

If $\|x - y\| > R$, then from $(\|x\| + \|y\|)^2 \leq 2\|x\|^2 + 2\|y\|^2$ it follows that

$$
\begin{aligned}
f(x, z) - f(y, z) &\leq M\Big(\sqrt{b/m}(\|x\| + \|y\|) + \|x\|^2 + \|y\|^2\Big) \\
&\leq \frac{M}{g(R)}g(\|x - y\|)\Big(1 + (b/m + 1)(\|x\|^2 + \|y\|^2)\Big) \\
&\leq \frac{M(b/m + 1)}{\varphi\varepsilon R}\rho(x, y).
\end{aligned}
$$

Combining these results, it follows that

$$f(x, z) - f(y, z) \leq \frac{M\big(b/m + 1\big)}{\varphi\varepsilon(R \vee 1)}\rho(x, y).$$

Now let $S, \widehat{S} \in \mathcal{Z}^n$ and suppose $\pi$ is the $W_\rho$-optimal coupling of $law(A(S))$ and $law(A(\widehat{S}))$ (which must exist due to Theorem 4.1 of [26]). If we consider the random variables $(X, Y) \sim \pi$, then it follows from above that

$$\mathbb{E}f(X, z) - \mathbb{E}f(Y, z) \leq \frac{M(b/m + 1)}{\varphi\varepsilon(R \vee 1)} W_\rho\Big(law(A(S)), law(A(\widehat{S}))\Big).$$

Since the right hand side is symmetric in $S$ and $\widehat{S}$, we find that this upper bounds $\varepsilon_{stab}(A)$. $\qquad\square$

In the proof of Theorem 3.1, we rely on the triangle inequality which is not available to us when using the metric $\rho$. However we can show a weak triangle inequality holds:

**Lemma D.1** (Weak triangle inequality). *For any $x, y, z \in \mathbb{R}^d$ it holds that,*

$$\rho(x, y) \leq \rho(x, z) + 2\Big(1 + \frac{R}{\varphi}(\varepsilon R \vee 1)\Big)\rho(z, y).$$

*Proof.* If both $\|x - z\|, \|z - y\| \geq R$, then the triangle inequality follows immediately from the definition of $\rho$:

$$\rho(x, y) = g(R)(1 + 2\varepsilon + \varepsilon\|x\|^2) + g(R)(\varepsilon\|y\|^2)$$
$$\leq \rho(x, z) + \rho(z, y).$$

In the case of $\|x - z\| \leq R, \|z - y\| \geq R$, we use the boundedness of $g$ as well as the inequality $\|x\|^2 \leq 2\|z\|^2 + 2R^2$ to deduce

$$\rho(x, y) \leq g(R)(1 + 2\varepsilon + \varepsilon\|y\|^2 + 2\varepsilon\|z\|^2 + 2\varepsilon R^2)$$
$$\leq 2(\varepsilon R^2 + 1)\rho(y, z).$$

For the final two cases, we first deduce the following inequality: if $\|x - y\| \leq R$, $\|x\|^2 - \|y\|^2 \leq \|x - y\|(\|x\| + \|y\|) \leq \frac{1}{\varphi}f(\|x - y\|)(\|x\| + \|y\|)$. From this, it follows that for the case $\|x - z\| \geq R, \|z - y\| \leq R$,

$$\rho(x, y) \leq g(R)(1 + 2\varepsilon + \varepsilon\|x\|^2 + \varepsilon\|z\|^2) + \varepsilon g(R)(\|y\|^2 - \|z\|^2)$$
$$\leq g(R)(1 + 2\varepsilon + \varepsilon\|x\|^2 + \varepsilon\|z\|^2) + \varepsilon\frac{g(R)}{\varphi}f(\|y - z\|)(\|y\| + \|z\|)$$
$$\leq \rho(x, z) + \frac{g(R)}{\varphi}\rho(y, z).$$

If $\|x - z\|, \|z - y\| \leq R$, we use the convexity of $g$ and the inequality $g(\|x - y\|) \leq g(\|x - z\|) + g(\|z - y\|)$ to deduce

$$\rho(x, y) \leq \rho(x, z) + \rho(z, y) + \varepsilon g(\|x - z\|)(\|y\|^2 - \|z\|^2) + \varepsilon g(\|z - y\|)(\|x\|^2 - \|z\|^2)$$
$$\leq \rho(x, z) + \rho(z, y) + \varepsilon\frac{1}{\varphi}g(\|x - z\|)g(\|z - y\|)(\|x\| + 2\|z\| + \|y\|)$$
$$\leq \rho(x, z) + \rho(z, y) + \varepsilon\frac{g(R)}{\varphi}g(\|z - y\|)(2R + 2\|z\| + 2\|y\|)$$
$$\leq \rho(x, z) + \Big(1 + \frac{2g(R)}{\varphi}(\varepsilon R \vee 1)\Big)\rho(z, y).$$

The result follows once it is noted that the coefficients derived in each case can be upper bounded by $2(1 + R(\varepsilon R \vee 1)/\varphi)$. $\qquad\square$

For computing the discretization error, we will find it easier to compute in the 2-Wasserstein distance, to this end we require the following lemma:

**Lemma D.2** (Comparison with the 2-Wasserstein distance). *For any two probability measures $\mu$ and $\nu$ on $\mathbb{R}^d$,*
$$W_\rho(\mu, \nu) \leq W_2(\mu, \nu)\big(1 + 2\varepsilon + \mu(\|\cdot\|^4)^{1/2} + \nu(\|\cdot\|^4)^{1/2}\big).$$

*Proof.* This follows immediately from the property $f(r) \leq r$ and the Cauchy-Schwarz inequality. $\qquad\square$

Finally, to compute the divergence bound, we will need the following result. Note that it is because of this result that we can only obtain $\mathcal{O}(\eta^{1/2})$ divergence bounds as oppose to the $\mathcal{O}(\eta)$ bounds obtained in the Lipschitz case.

**Lemma D.3.** *Suppose $X, Y, \Delta_x$ and $\Delta_y$ are random variables on $\mathbb{R}^d$, then*

$$\mathbb{E}\rho(X + \Delta_x, Y + \Delta_y) \leq \mathbb{E}\rho(X, Y) + \sigma_\Delta^{1/2}(1 + 2\varepsilon + 6\varepsilon\sigma^{1/2}),$$

*where we define $\sigma_\Delta := \mathbb{E}\|\Delta_x\|^2 \vee \mathbb{E}\|\Delta_y\|^2$ and $\sigma := \mathbb{E}\|X\|^4 \vee \mathbb{E}\|Y\|^4 \vee \mathbb{E}\|X + \Delta_x\|^4 \vee \mathbb{E}\|Y + \Delta_y\|^4$.*

*Proof.* Using the convexity of $g$ which yields the inequality $g(\|X + \Delta_x - Y - \Delta_y\|) \leq g(\|X - Y\|) + g(\|\Delta_x - \Delta_y\|)$, as well as the property $g(r) \leq r$, it follows that

$$\begin{aligned}
\rho(X + \Delta_x, Y + \Delta_y) \leq{}& \rho(X, Y) + \varepsilon g(\|X - Y\|)(\|X + \Delta_x\|^2 - \|X\|^2 + \|Y + \Delta_y\|^2 - \|Y\|^2)) \\
&+ g(\|\Delta_x - \Delta_y\|)(1 + 2\varepsilon + \varepsilon\|X + \Delta_x\|^2 + \varepsilon\|Y + \Delta_y\|^2) \\
\leq{}& \rho(X, Y) + \varepsilon\|\Delta_x\|(\|X\| + \|Y\|)(\|X\| + \|X + \Delta_x\|) \\
&+ \varepsilon\|\Delta_y\|(\|X\| + \|Y\|)(\|Y\| + \|Y + \Delta_y\|) \\
&+ (\|\Delta_x\| + \|\Delta_y\|)(1 + 2\varepsilon + \varepsilon\|X + \Delta_x\|^2 + \varepsilon\|Y + \Delta_y\|^2).
\end{aligned}$$

For any three random variables $A, B, C$, the Cauchy-Schwarz inequality can be applied twice to obtain $\mathbb{E}ABC \leq (\mathbb{E}A^2)^{1/2}(\mathbb{E}B^4)^{1/4}(\mathbb{E}C^4)^{1/4}$. From this we deduce the following:

$$\begin{aligned}
\mathbb{E}\rho(X + \Delta_x, Y + \Delta_y) \leq{}& \mathbb{E}\rho(X, Y) + \varepsilon\sigma_\Delta^{1/2}(2\sigma^{1/4})(2\sigma^{1/4}) + \varepsilon\sigma_\Delta^{1/2}(2\sigma^{1/4})(2\sigma^{1/4}) \\
&+ 2\sigma_\Delta^{1/2}(1 + 2\varepsilon + \varepsilon\sigma^{1/2} + \varepsilon\sigma^{1/2}) \\
\leq{}& \mathbb{E}\rho(X, Y) + 2\sigma_\Delta^{1/2}(1 + 2\varepsilon + 6\varepsilon\sigma^{1/2}).
\end{aligned}$$

$\square$

## D.2 Proof of Theorem 4.1

We give a similar argument to that given in the proof of Theorem 3.1. Using the property of the semimetric given in Lemma D.3 as well as the results of Lemmas B.3 and A.1, it follows that

$$\begin{aligned}
W_\rho\left(\mu P_\eta^B, \nu \widehat{P}_\eta^B\right) \leq{}& W_\rho(\mu, \nu) + 2\eta^{1/2}\left(M^2\sigma_\Delta^{1/2} + M^2\frac{3b + 2d/\beta}{m} + d/\beta\right)^{1/2} \\
&\cdot \left(1 + 2\varepsilon + 6\varepsilon\sigma_\Delta^{1/2} + 12\varepsilon[b/m + (d + 2)/\beta m]\right),
\end{aligned}$$

where $\sigma_\Delta = \mu(\|\cdot\|^4) \wedge \nu(\|\cdot\|^4)$. Furthermore, if we suppose that $\mu = \mu_0 R_X^t$ and $\nu = \mu_0 \widehat{R}_X^t$ for some $t$, then with Lemma B.2 we obtain $\sigma_\Delta \leq \rho_0(\|\cdot\|^4) + \tilde{c}(2)$ (see Lemma B.2 for the definition of $\tilde{c}(p)$) and thus we obtain the bound $W_\rho\left(\mu P_\eta^B, \nu \widehat{P}_\eta^B\right) \leq W_\rho(\mu, \nu) + \tilde{c}_2\eta^{1/2}$ with constant

$$\begin{aligned}
\tilde{c}_2 := {}& 2\left(M^2\sigma_4^{1/2} + M^2\tilde{c}(2)^{1/2} + M^2\frac{3b + 2d/\beta}{m} + d/\beta\right) \\
&\cdot \left(1 + 2\varepsilon + 6\varepsilon\sigma_4^{1/2} + 6\varepsilon\tilde{c}(2)^{1/2} + 12\varepsilon[b/m + (d + 2)/\beta m]\right).
\end{aligned}$$

Borrowing the convexity argument given in the proof of Theorem 3.1, it follows from the contraction in Lemma C.1 and the above equation that

$$W_\rho\left(\mu R_X, \nu \widehat{R}_X\right) \leq \tilde{c}_3\, W_\rho(\mu, \nu) + \tilde{c}_2\frac{k}{n}\eta^{1/2},$$

where $\tilde{c}_3 := \frac{k}{n} + (1 - \frac{k}{n})e^{-\eta/C_4}$. Thus it follows by induction that

$$W_\rho\left(\mu_0 R_X^t, \mu_0 \widehat{R}_X^t\right) \leq \frac{1 - \tilde{c}_3^t}{1 - \tilde{c}_3}\tilde{c}_2\frac{k}{n}\eta^{1/2}.$$

Using Lemma 4.3, it follows that

$$\varepsilon_{stab}(X_{\eta t}) \leq C_5\frac{1 - \tilde{c}_3^t}{1 - \tilde{c}_3}\frac{k}{n}\eta^{1/2},$$

with constant,

$$C_5 := \frac{M\left(\sigma_2 + \sqrt{b/m}\right)}{\varphi\varepsilon(R \vee 2)}\tilde{c}_2.$$ (12)

After recalling the argument from the proof of Theorem 3.1 that deduces $\frac{1-\tilde{c}_3^t}{1-\tilde{c}_3} \leq \min\{t, \eta(C_4 + 1)/(1 - k/n)\}$, the bound in the statement follows.

Next, from the weak triangle inequality in Lemma D.1 it follows that,

$$\mathrm{W}_\rho\left(\mu_0 R_x^t, \mu_0 \widehat{R}_x^t\right) \leq \tilde{c}_4 \, \mathrm{W}_\rho\left(\mu_0 R_x^t, \mu_0 R_X^t\right) + \mathrm{W}_\rho\left(\mu_0 R_X^t, \mu_0 \widehat{R}_X^t\right) + \tilde{c}_4 \, \mathrm{W}_\rho\left(\mu_0 \widehat{R}_X^t, \mu_0 \widehat{R}_x^t\right),$$

with $\tilde{c}_4 := 1 + \frac{2g(R)}{\varphi}(\varepsilon R \vee 1)$. Using the comparison with the 2-Wasserstein distance in Lemma D.2 and the bounds in Lemmas B.4 and B.2 we obtain $\mathrm{W}_\rho(\mu_0 R_x^t, \mu_0 R_X^t) \leq \tilde{c}_5\eta^{3/2}$, where

$$\tilde{c}_5 := 2\sqrt{2}\exp(M^2)M\left(M^2\sigma_4^{1/2}+M^2\tilde{c}(2)^{1/2}+M^2b/m+\beta^{-1}d\right)^{1/2}\left(1+2\varepsilon\left[1+\sigma_4^{1/2}+\tilde{c}(2)^{1/2}\right]\right).$$

Thus, we can compute another contraction estimate and using Lemma 4.3

$$\varepsilon_{stab}(x_t) \leq C_6 \frac{1 - \tilde{c}_2^t}{1 - \tilde{c}_2}\left(\frac{k}{n}\eta^{1/2} + \eta^{3/2}\right),$$

$$C_6 := \frac{M\left(\sigma_2 + \sqrt{b/m}\right)}{\varphi\varepsilon(R \vee 2)}(\tilde{c}_2 \vee 2\tilde{c}_4\tilde{c}_5).$$ (13)

$\square$