# OpenReview forum: "Time-independent Generalization Bounds for SGLD in Non-convex Settings"
_NeurIPS.cc/2021/Conference — NeurIPS 2021 Poster_

### Official Review · Reviewer_Ki1F · 2021-06-27

**Rating:** 7
**Confidence:** 4

**Summary:**

In this paper, the authors investigate the stability of stochastic gradient Langevin dynamics (SGLD) with varying data sets. They prove a time-independent stability bound for SGLD in non-convex settings. And their analysis can also be used to analyze other variants of SGLD.

More specifically, the reviewer summarizes their contribution as follows: In this paper, they bound the generalization error to show the stability of SGLD. As claimed by the authors, unlike existing bounds for SGLD in non-convex setting, their bound is time-independent and decays to zero as data size increases.


**Ethical Concerns:**

I think there aren't any ethical issues with this paper.

**Limitations And Societal Impact:**

There are several limitations and concerns that the reviewer wants to mention:
1. The authors mention they require the learning rate to be sufficiently large relative to n. This is counter-intuitive. The reviewer thinks if \eta is small enough, then the whole system will converge to an SDE. And it's often easier to study the stability of SDE. Can the authors give more comments on this?
2. In Theorem 3.1 and 4.1, the authors give an equal requirement for time step $\eta$ in the discretization case. This seems too strict. Is it possible to give a looser requirement for $\eta$?
3. The authors need a weight term in their result, is this purely technical?
4. The reviewer quickly went through the proof of Proposition 4.2 (Appendix E.1). The reviewer is curious about how the authors get the second inequality in (18).  It seems the authors use the contraction property of Langevin dynamics (Lemma 3.4). But the assumptions of Lemma 3.4 are no longer true in Section 4. The reviewer might miss something.
5. Although the authors give an improved result, the paper lacks applications. Is it possible to give some applications?

There are some structural problems and typos that the reviewer wants to mention:
1. The authors put the definition of $\bar{F}_S$ at the beginning of Section 3. This is weird since this definition is also used in Section 4.
2. In Section 4, $m$ is abused. It is used for both dissipative constant and iteration steps.
3. There are a lot of parameters in this paper. The authors might recall the origin of some parameters when they use them. One example is $\lambda$ in Theorem 3.1 and 4.1.
4. When the authors mention continuous-time algorithm or discrete-time algorithm, the authors can also cite the equations. Then it will be more clear to readers.

**Main Review:**

Quality, significance: Overall, the reviewer thinks this is a good paper.  They give a time-independent bound for generalization error in non-convex setting. There haven't been any results like this before. (This is claimed by the author, in fact, the reviewer is not familiar with the reference in this area.) The reviewer thinks this is an interesting and reasonable improvement.

Clarity: On the whole, the review thinks the organization of this paper is clear. But there are still some misleading parts, the review will mention them in "Limitations".

Originality:  The review thinks this paper is novel enough to be published. The reviewer has done some research in Langevin dynamics before. So the reviewer thinks that the idea in this paper is not very surprising and creative. But it's good that the author can combine different results together to bound the generalization error.

**Time Spent Reviewing:**

3 hours

---

> ### Author Response · Authors · 2021-08-10
> **Author response to reviewer Ki1F**
>
> We thank reviewer Ki1F for their valuable comments and feedback.
>
> ## D.1 – Needing $\eta$ to be sufficiently large is counter-intuitive.
> We agree with the reviewer that when $\eta$ is taken to be smaller, SGLD more accurately approximates an SDE. In fact, it has been shown that under assumptions 4.1-4.3, as $\eta \to 0$, SGLD converges uniformly over time in Wasserstein distance to the SDE defined below line 121 -- the Langevin equation for the full batch empirical risk [5, 31]. However, since the SDE is defined using the full-batch empirical risk and not the mini-batch average, analyzing the stability of this SDE under the assumptions of Section 4 does not appear to be an easy task. The methods available for estimating the distance between two SDEs with different drifts seem of limited use in this respect, and it no longer seems possible to establish a contraction, at least with the methods we are aware of.
>
> It is indeed the case that a smaller discretization error would improve the generalization bound, but the learning rate not only determines the size of the discretization error, it also determines how fast the mini-batch is updated. As a result, both the contractive property and the stability induced by the use of stochastic gradients are only noticeable for larger $\eta$. In Remark 3.7, we separate these two dependencies on the learning rate by updating the random mini-batch only every $\lfloor 1/\eta \rfloor$ iterations which results in better generalization bounds.
>
> Our intuition for the assumption placed on $\eta$ as well as its purpose in the proof is discussed in our response to reviewer v2Zx, Section B.3.
>
>
> ## D.2 – Is it possible to give looser requirements for $\eta$ in the discrete case?
> In Theorems 3.1 and 4.1, the learning rate is written in terms of the parameter $\alpha$ to enforce the optimal scaling of $\eta$ with respect to $k$ and $n$. In Theorem 3.1 this is written as $\eta = \alpha k/n$ for $\alpha \geq C_1$, for instance. Though the fact that this assumption is written as an equality may make it seem restrictive at first glance, $\eta$ is still free to take any value that it could take in the respective continuous-time case. In Propositions 3.2 and 4.2, we detail how the discrete-time bound scales with $\eta$.
>
>
> ## D.3 – Is weight decay necessary?
> We refer the reviewer to our discussion in our response to reviewer hd87, Section A.4.
>
>
> ## D.4 – How does equation (18) come about?
> Lemma C.1 in the Appendix, shows that a contraction holds under Assumptions 4.1 and 4.2. Equation (18) is formed by combining this Lemma with (17) using the convexity of the Wasserstein distance.
>
>
> ## D.5 – The authors put the definition of $\tilde{F}_S$ at the beginning of Section 3.
> The reviewer's concern regarding the placement of the definition of $\tilde{F}_S$ suggests that there may be a misunderstanding of the setting of Theorem 4.1. The setting of Theorem 4.1 is that SGLD is performed with a smooth dissipative loss function without weight decay. It is only in Section 3 that we consider the setting of Lipschitz loss functions with weight decay, which is a special case of the setting in Theorem 4.1. We hope this addresses the reviewer's concern.
>
>
> ## D.6 – The paper lacks applications.
> Our results consider the setting of dissipative smooth loss functions. We refer the reviewer to our response to reviewer WU19, Section C.3, as we highlight how this setting is known to apply to a variety of practical settings. Furthermore, we discuss the consequences of our bounds in our response to reviewer hd87, Section A.1.

---

> > ### Comment · Reviewer_Ki1F · 2021-08-11
> > **Response to the authors**
> >
> > The reviewer is satisfied with the authors' response. The reviewer thinks the authors have addressed all the concerns.

---

### Official Review · Reviewer_WU19 · 2021-07-14

**Rating:** 7
**Confidence:** 3

**Summary:**

The paper proves generalization bounds for stochastic gradient Langevin dynamics (SGLD) for nonconvex functions. Previous bounds for SGLD in similar settings typically goes to infinity as time or the number of steps increases. This work gives bounds that are independent of time, and also decreases to 0 as the sample size $n$ grows.

Similar to previous bounds, the authors prove that SGLD is uniformly stable, which is known to imply good generalization. The proofs of stability proceed by starting with the continuous version of the algorithm, and then apply discretization techniques to get guarantees for the discrete-time SGLD. This discretization leads to a slower rate: the bounds are $O(1/n)$ vs $O(\sqrt{d/n})$ for Lipschitz loss functions plus weight decay, and $O(1/n^{1/3})$ vs $O(1/n^{1/6})$ for the full dissipative case without the Lipschitz-ness assumption.

For Lipschitz loss functions, it is sufficient to control the 1-Wasserstein distance ($W_1$) between the distributions induced by two neighbouring datasets. The nontrivial step is Lemma 3.4, which shows that conditioning that the differing data point, $n$, is not in the minibatch, some proxy for $W_1$, $W_g$, actually gets contracted due to the regularization term. Then, as long as the minibatch size is relatively small, the overall distance also gets shrinked. This leads to the time-independent bound on stability.

The proof becomes more challenging without the Lipschitz-ness assumption, since controlling $W_1$ alone does not guarantee stability. The authors show that controlling $W_2$ instead would work, though the contraction proof becomes more involved.

**Limitations And Societal Impact:**

Adequately addressed.

**Main Review:**

I found the paper very well-written, as it includes sufficient motivation and background for the problem, and the main paper contains an appropriate amount of technical details to allow the readers to understand the general proof strategy.

I have two questions:
- Theorem 3.1: The precondition requires the minibatch size, $k$, to be relatively small to the dataset size $n$. In particular, the theorem does not seem to hold for the full gradient setting $k = n$, whereas the result of Li et al. [16] (only) holds for this setting. Some discussion about this distinction could be helpful.
- Line 298: Is $\mathbb{E}\|A(S)\|^2 \le s^2$ an additional assumption for Theorem 4.1 to hold (instead of something that can be proven)? If so, it should also appear in the theorem statement.

The analyses follow a similar roadmap to those of previous stability bounds, though a few new tools (like considering both $W_1$, $W_2$ and their proxies under different settings) are needed to establish the required contraction phenomena and time-independent bounds. My only reservation is that the applicability of the results to actual ML settings, since they either require $d \ll n$ (which is not true for overparametrized models) or the dissipativity condition which is too strong to hold for real-world loss functions. That said, the current paper already contains interesting new results to a problem of much recent attention, so I recommend accept.

===Added After Author Response===

I have read the authors' response, and would like to thank the authors for answering the questions. My overall evaluation of the paper stays positive.

**Time Spent Reviewing:**

2

---

> ### Author Response · Authors · 2021-08-10
> **Author response to reviewer WU19**
>
> We thank reviewer WU19 for their valuable comments and feedback.
>
> ## C.1 – Why does the analysis require $k < n$
> The reviewer is correct, our result does not apply to the case where $k=n$ and in fact it is designed for the case $k \ll n$. Our proof relies heavily on the contractive property of SDEs which is only possible to establish, to the best of our knowledge, when the two SDEs share the same objective function/bias term. For instance, in the case where $k=n$ the full-batch averages $F_S$ and $F_{\hat{S}}$ may not be equal and so such a result is not necessarily possible to establish. With $k < n$ there is a $1-k/n$ probability that $F_S(\cdot, B) = F_{\hat{S}}(\cdot, B)$ and so the Langevin diffusions based on these functions can indeed contract.
>
>
> ## C.2 – Is $\mathbb{E}|A(S)| \leq s^2$ an additional assumption?
> No, the assumptions discussed on line 297 are those required for Lemma E.1 to hold. All assumptions required for Theorem 4.1 are established in assumptions 4.1-4.3 and partly in Section 2. We show in Lemma B.2 that under assumptions 4.1-4.3, SGLD has finite second moment.
>
>
> ## C.3 – The results are not applicable since dissipativity is too stronger assumption.
> The reviewer claims that the assumption of dissipativity does not hold for real-world loss functions. As outlined in our paper, the assumption of dissipativity has been considered by a variety of works on SGLD, including the seminal paper [22] by Raginsky et al. In [31], Zhang et al. show that an example of variational inference for Bayesian logistic regression is dissipative and smooth. In [5], Chau et al. show that any learning problem constrained to a Euclidean ball with a continuously differentiable loss function can be made into a dissipative learning problem on the full Euclidean space. Furthermore, they show that their analysis of SGLD can be applied to the training of feed-forward neural networks.
>
> In Section 3, we consider the most simple dissipative objective function: a Lipschitz loss function with weight decay regularization. Further, we remark that any smooth (in the sense of assumption 3.2/4.2) loss function with weight decay regularization satisfies this dissipativity assumption.
>
>
> ## C.4 – Theorem 3.1 is not applicable in the high-dimensional setting.
> The reviewer's concerns about Theorem 3.1 not supporting the high-dimensional setting have been addressed in our response to reviewer v2Zx in Section B.7. In fact, the dimension-free nature of the bounds in Theorem 3.1 suggest SGLD may be a promising algorithm for high-dimensional setting.

---

### Official Review · Reviewer_v2Zx · 2021-07-16

**Rating:** 6
**Confidence:** 3

**Summary:**

This paper proves generalization bounds for SGLD under dissipativity and smoothness assumptions. A nice feature of these bounds is that they are "time-independent" and uniformly hold across all iterations.

**Main Review:**

*Main Comments*
- The paper requires a more detailed comparison with previous work and a better interpretation/justification for the bounds (see below).
- Although the "time-independent" bounds are elegant, and the analysis is novel, I do not understand the utility of these bounds even in theoretically understanding the fundamental dependencies on the dimension, batch-size or the step-size in the discrete setting. In these terms, the paper needs to be better motivated for the machine learning audience.

*Details*
- In Theorem 3.1, the constants $C_1, C_2, C_3$ depend exponentially on the $M, L, \beta, \lambda$. Is there a lower bound that justifies this bad dependence?
- In this theorem, please explain the effect of the regularization - how does it help the generalization (does the bound capture the difference between the regularized case, and when $\lambda = 0$)? SImilarly, what is the effect of the batch-size $k$ (do we expect better generalization bounds for smaller batch-sizes)? What is the intuition behind the $\eta \geq C_1 \frac{k}{n}$ condition needed in the theorem?
- It would be helpful to compare Theorem 3.1 to the bounds obtained in the previous work, for example, [22]
- Similarly, for Theorem 4.1, please compare the obtained bounds to the previous work. And are there corresponding lower bounds to explain the dependence on $n$, and the exponential dependence on the constants? It is unclear to me which dependencies are fundamental vs those that are limitations of the analysis technique.
- Comparing the bounds in Theorems 3.1 and 4.1, with the Lipschitz assumption, the bounds for the discrete case are better in $n$, but suffer a dimension dependence. For Thm 4.1, the discrete case bounds have worse dependence on $n$ which makes sense, but they do not have a dimension dependence. What is the reasoning behind this?
-




**Time Spent Reviewing:**

5

---

> ### Author Response · Authors · 2021-08-10
> **Author response to reviewer v2Zx**
>
> We thank reviewer v2Zx for their valuable comments and feedback.
>
> ## B.1 – I do not understand the utility of these bounds even in theoretically understanding the fundamental dependencies...
> We refer the reviewer to the response to reviewer hd87, Section A.1 and Section A.2.
>
>
> ## B.2 – Please explain the effect of the regularization - how does it help the generalization?
> In the case that $\lambda \leq M$, the inverse of the contraction coefficient $1/c$ is of the form $\exp(C (M-\lambda)/\lambda^2)(1/\lambda^2 + 1/\lambda)$ where $C$ is constant with respect to $M$ and $\lambda$. Thus, as $\lambda$ approaches $M$ from below, $1/c$ decays at a rate of $1/\lambda$. In the case that $\lambda \geq M$, $1/c$ is constant with respect to $M$ and has rate $1/\lambda$. The term $M-\lambda$ also appears in the results of Hardt et al. [13], who show that hyperparameter $\lambda$ counters and reduces the smoothness coefficient.
>
> However, since the discretization error worsens exponentially as $\lambda$ grows, the final generalization error bound has rate $\exp(\lambda + M)/\lambda$ as $\lambda$ approaches $M$ from below or for $\lambda \geq M$. This is a strongly-convex function of $\lambda$ with its minimizing value in $\mathbb{R}^+$.
>
> Our bounds may not shrink as $\lambda$ increases but we do find that in the setting of Section 3, weight decay is instrumental in computing time-independent generalization error bounds. Indeed, it is by adding weight decay that the objective function becomes confining/dissipative which is what allows us to observe contractions. This is demonstrated in the way that our bounds diverge as $\lambda \to 0$.
>
>
> ## B.3 – What is the intuition behind the $\eta \geq C_1 k/n$ condition?
> The ratio between the learning rate and batch size has previously been connected to the generalization capabilities of SGMs as a result of empirical results. For example, Jastrzębski et al. [*] provide empirical evidence that the generalization capabilities of SGD are determined by the ratio $\eta / k$. In particular, they show that as $\eta / k$ decreases, validation accuracy worsens. With this, our assumptions on the size of $n$ are unsurprising: if $\eta / k$ is made smaller we require more data to observe good generalization.
>
> In our proof, this assumption is fundamental. At each iteration our coupling $(X_t, \hat{X}_t)$ can either contract in Wasserstein distance at the rate $e^{-O(\eta)}$ or, if the differing data point is in the mini-batch, it diverges. Therefore, we need $\eta$ large enough so that the contractions make up for the iterations in which the coupling diverges. Since $k/n$ is the probability of this data point being in the mini-batch, $\eta$ has to grow relative to this quantity.
>
>
> ## B.4 – Is there a lower bound that justifies this bad dependence [on the hyperparameters]?
> We are not aware of any lower bounds for the generalization error of SGLD or statistical lower bounds for the case of dissipative loss functions.
>
>
> ## B.5 – Please compare the obtained bounds to the previous work
> Mou et al. [17] show that under the assumption that the loss function is Lipschitz and bounded, the expected generalization error after $T$ iterations is bounded by $O(\sqrt{\eta T}/n)$. This was made more general by Pensia et al. [21] who, under the assumption that the loss function is Lipschitz and subgaussian, obtained the bound $O(\sqrt{\eta T/n})$.
>
> To the best of our knowledge, our results are the only data-independent bounds under the assumption of dissipativity. Though they do not explicitly aim to devise generalization error bounds, under the same assumptions as in Section 4, Raginsky et al. [22] obtain the expected generalization error bound $O(T\eta + e^{-cT\eta} + 1/n)$
>
> There are a variety of data-dependent results that recover the $\sqrt{\eta/n}$ scaling with fewer assumptions on the loss function [17, 18, 16, 12]. However, the bounds depend on the expected norm of the gradient at each iteration and if we wish to use a Lipschitz assumption to bound these quantities, we recover the previously mentioned results of [17] and [21]. In particular, Theorem 10 of Mou et al. [17] produces data-dependent bounds for the case of Lipschitz loss functions with weight decay regularization, the same setting as Section 3.
>
> In the setting of Section 3, our bounds have the same $O(1/\sqrt{n})$ rate but, unlike Theorem 10 of Mou et al., our bounds are guaranteed to not explode with time, are not data-dependent, and do not require the loss function to be subgaussian. In Remarks 3.6 and 4.3 we discuss how the bounds grow with rate $\eta T$ in the setting of Section 3 and with rate $\sqrt{\eta T}$ in the setting of Section 4 when the number of iterations $T \ll 1/\eta$.
>
>
> ## B.6 – It is unclear to me which dependencies are fundamental vs those that are limitations of the analysis technique.
> The fundamental message of the analysis is that SGLD has the ability to generalize effectively as a result of two fundamental properties: the contraction phenomenon of SDEs and the stability induced by the use of stochastic gradients. The assumption of dissipativity in Theorem 4.1 and the use of weight decay in Theorem 3.1 is what facilitates the contractive property. As we discussed in Section B.3 of this comment, having $\eta$ grow with $k/n$ is also fundamental in facilitating the combination of these two properties.
>
> Assumptions 3.3 and 4.3 are technicalities that guarantee that our iterate is integrable in some sense. The assumption of smoothness allows for the discretization error to be uniformly bounded and guarantees the existence of a strong solution to the SDE in (2) (see Theorem 3.1 of [20]).
>
>
> ## B.7 – With the Lipschitz assumption the bounds are worse in dimension than the more general case. Why?
> The reviewer is commenting on the fact that in Theorem 4.1, the bounds only depend on $\beta$ and $d$ through the ratio $\beta / d$ and so the bounds are dimension-free when $\beta \propto d$. Furthermore, in Theorem 3.1 the continuous-time bound depends exponentially on $\beta$ and not $d$ and so when the $O(1 + d/\beta)$ discretization error is introduced in the discrete-time setting, it is not possible to enjoy the same dimension-free qualities as in the more general case.
>
> We agree with the reviewer that this is undesirable and rules out the application of Theorem 3.1 to the high-dimensional setting. As a result, we have made a small change to the proof of Lemma 3.4 (Appendix C) so that $c$ and $c_1$ are independent of $\beta$ and $d$ and, as a result, Theorem 3.1 is now dimension-free when $\beta \propto d$.
>
> The change to the proof is as follows: recall that in Appendix C.1 we want to control the quantity $r_t = \|X_t - Y_t\|$. Since $\beta^{-1} \kappa(r)$ is constant with respect to $\beta$ we can apply the method of Eberle [8], as outlined on page 18, to obtain a function $g$ that satisfies $g''(r) - \frac{1}{4}r \beta^{-1} \kappa(r) g'(r) \leq - \frac{\hat{c}}{2} g(r)$ with $\hat{c}$ and $g(r)$ constant with respect to $\beta, d$. Additionally, for some $p > 0$ we can define the function $\tilde{g}(r) = p (r - R) \vee 0$ which for any $r \neq R$, satisfies $- \frac{1}{4} r \beta^{-1} \kappa(r) \tilde{g}'(r) \leq - \beta^{-1} b \tilde{g}(r)/4$. Now if we assume that $\beta \geq 2$ we can define the function $\hat{g}(r) = 2\beta^{-1} g(r) + (1 - 2\beta^{-1}) \tilde{g}(r)$ which for $r \neq R$, satisfies the equation at the end of line 598 with $c = \min\{\hat{c}, \lambda / 4\}$. Following from the argument up to line 600, we obtain the contraction $\mathbb{E} \hat{g}(r_t) \leq e^{-ct} \mathbb{E} \hat{g}(r_0)$. Since, by line 604 we have the bound $\varphi_{min}/2 \leq g' \leq 1$ for some $\varphi_{min}$ that is independent of $\beta$, we can also deduce that $\mathbb{E} g(r_t) \leq (\beta/2) e^{-ct} \mathbb{E} \hat{g}(r_0) \leq e^{-ct} \mathbb{E} g(r_0) \big ( 1 + p (\beta - 2)/\varphi_{min} \big )$. Since $p$ can be made arbitrarily small, we obtain the contraction $\mathbb{E} g(r_t) < e^{-ct} \mathbb{E} g(r_0)$.
>
> We thank the reviewer for pointing this out and giving us the chance to implement this simple fix.
>
> ## References
>
> [*] Jastrzębski S, Kenton Z, Arpit D, Ballas N, Fischer A, Bengio Y, Storkey A. Three factors influencing minima in sgd. arXiv preprint arXiv:1711.04623. 2017 Nov 13.

---

> ### Author Response · Authors · 2021-08-29
> **Correction to Author Response**
>
> We are following-up on point B.7 in our previous reply, where we outlined a sketch of a modification of the proof of Theorem 3.1 so that the bound can be made dimension-free, matching the dimension-free nature of Theorem 4.1 when $\beta \propto d$.
>
> While working on the details to include the final result in our paper, we have now noticed that there seems to be a problem with our proposed strategy to modify the proof and that the dimension-dependence nature of that result in our submission appears to be fundamentally linked to the approach that Eberle [8] uses to attain the $W_1$ contraction, which is what allowed us to establish $1/\sqrt{n}$ rates (one of the main objectives of our work is to establish time-independent bounds that go do zero as $n$ increases). We are exploring this more in depth, and will include an extensive comment on this in the camera-ready version of our work.
>
> Nevertheless, within the context of our setting and algorithm, we believe the original result as presented in our submission is already of interest, despite its dependence on $d$. We have also included this result in the paper for two additional reasons. Firstly, because it helps us establish the methodology used in a way that is simpler than the more general case. Secondly, because we obtain better rates w.r.t. n; this is at the expense of bad dimensional dependence, at least in this case.

---

> ### Comment · Reviewer_v2Zx · 2021-08-30
> **Response to authors**
>
> I thank the authors for their detailed response, and their attempt to improve the bounds in Section 3. I have read through the other reviews and the author's responses.
>
> The merits of the paper are (i) the dimension-independent, time-independent, data-independent bounds without the Lipschitz assumption (Section 4) are novel, and (ii) recovering the $k/n$ dependence for the step-size $\eta$ is interesting. While it is unfortunate that dimension-independent bounds for Sec 3 do not seem immediately possible, the authors have provided a clearer comparison of their bounds to the previous work and addressed my other concerns. I'm satisfied with their response.
>
> I will increase my score from 5->6.
>
> I request the authors to use the rebuttal to clearly motivate their results, add a more detailed discussion comparing to the previous work and situating their work, and clearly explain the downsides of their analysis technique that lead to the dimension-dependence in Section 3. It would also be helpful to explicitly differentiate between the dependencies (on $n$, $\lambda$) that the authors think are fundamental, while which of those ($d$ in Sec 3) are a limitation of the analysis techniques.

---

### Official Review · Reviewer_hd8z · 2021-07-16

**Rating:** 5
**Confidence:** 2

**Summary:**

The paper derives time-independent generalization bounds for (discrete- and continuous- time) SGLD under dissipativity and smoothness assumptions. The main novelty of the new bounds appears to be the fact that they decay to zero as the sample size increases. The authors claim that the theoretical approach is different to prior related works on SGLD as it relies on the framework of uniform stability.

**Limitations And Societal Impact:**

I would like to see more discussion on the consequences of the new bounds in the practice of non-convex optimization. See above.

**Main Review:**

Before everything, let me say that I am not quite familiar with this line of work on analyzing SGLD in non-convex settings. Thus, even though I have not found a technical issue in the proof sketch in the main body, I cannot guarantee correctness of the all the proofs.

In general, the paper is well written with only a few typos. Main results and accompanying assumptions are clearly stated. Also, comparison to related literature is thorough and seems satisfactory, although at certain places it is mostly targeting the expert reader rather than the general Neurips audience.

I understand from the presentation that the results are novel and the technique differs from previous most closely related works. In particular, it is claimed that the authors are the first to bring the contraction property of Langevin dynamics in Wasserstein distance from the optimization/sampling literature to generalization analysis.

However, I am not convinced about the consequences of the new bounds. What do we learn from them about SGLD behavior (that is perhaps surprising or new)? Are there any implications ---even at a high-level--- about SGD schemes used in practice of non-convex optimization? Does it say something about how to choose the learning parameter? Or under what conditions on loss and data is generalization good?

I would like to see more such discussion and insights on the bounds that would also benefit the broader Neurips crowd.

Other questions:
1. Why are *time-independent* bounds desirable? Wouldn't it be better that they depend on time so that they are informative about the algorithm's evolution over time. This seems to not be explained in the paper.

2. Lines 84-85: How do your assumptions on the learning rate relate to practical choices?

3. Equation end of page 4: Why is the weight-decay in the form of \ell_2, rather than \ell_2^2 norm? Also, is the weight decay needed here for the analysis?

**Time Spent Reviewing:**

2

---

> ### Author Response · Authors · 2021-08-10
> **Author response to reviewer hd8z**
>
> We thank reviewer hd87 for their valuable comments and feedback.
>
>
> ## A.1 – I am not convinced about the consequences of the new bounds.
> As we have outlined in our paper, there is a long and ongoing literature establishing different types of generalization bounds for computationally efficient first-order methods, and our work sits within this literature. We show that SGLD has the ability to generalize effectively as a result of two fundamental properties: the contraction phenomenon of SDEs and the stability induced by the use of stochastic gradients. Our results suggest that to guarantee that SGLD generalizes well we should consider settings and select hyperparameters that facilitate both of these properties.
>
> Some immediate practical insight to come of this is that there may be some benefit, in terms of generalization capabilities, to *not* setting $\eta$ as small as possible. Indeed, by requiring $\eta$ to be larger than some function of $n$ we facilitate contractions that lead to generalization bounds that don't explode with time (we highlight this on lines 84 and 321). In the seminal paper [13], Hardt et al. claim that training faster, i.e. having $\eta$ and $T$ small, leads to better generalization for SGD, and existing generalization bounds for SGLD echo this sentiment [17, 21]. Our results extend upon these claims by showing that when $T$ is large we can still generalize well if $\eta$ is taken sufficiently large relative to $1/n$.
>
> Also, for the contraction to take place it is required that the empirical risk is confining in some sense (e.g. dissipative or using weight decay). Our results suggest that SGLD generalizes effectively in these settings.
>
> Our proof technique allows for some modifications to the SGLD algorithm and we discuss ways in which SGLD can be improved. In Remark 3.7, we show how it is possible to obtain faster $O(1/n)$ generalization bounds by updating the random mini-batch only every $\lfloor 1/\eta \rfloor$ iterations. In Remarks 3.8 and 3.9, we discuss how the same bounds hold for projected SGLD and for SGLD with non-isotropic Gaussian noise, suggesting that the generalization capabilities of SGLD are not strictly connected to the ergodic property that arises in the case of isotropic noise without projections.
>
>
> ## A.2 – Why are time-independent bounds desirable? Wouldn't it be better that they depend on time...
> Emphasis on establishing time-independent bounds is already given in the seminal paper [13] by Hardt et al., for instance, where they establish it for SGD in strongly-convex settings. Our paper seems to be the first to establish analogous theoretical results for SGLD, which is receiving increasing interest in the learning community. For SGLD, empirical findings have shown that, in some settings, the generalization error does not explode with time (see for example Figure 1 of [3] and Figure 1 of [18]). To the best of our knowledge, prior to our work, this observation has not been demonstrated in any theoretical results outside of the strongly-convex setting (c.f. discussion below). The authors of [3, 18] motivate the use of data-dependent bounds precisely because the data-independent bounds of Mou et al. [17] and Pensia et al. [21] overestimate the growth of these bounds over time.
>
> For SGD, existing works have devised bounds that do not explode with time in the strongly convex setting [13, 6] and there has been work that extends this to more general settings (see Mingyang Yi et al. [**], for instance). Despite growing interest in this topic, we are not aware of any analysis that shows that, in a non-convex setting, an algorithm achieves time-independent generalization bounds whilst retaining a bound that decreases to $0$ as $n$ increases; this is the flavor of some results in [13], just to mention an example. To the best of our knowledge, we are the first to demonstrate that this property holds, in the case of SGLD.
>
> We agree with the author that it may be desirable to know the rate at which the error grows with time. For that reason, in Remarks 3.6 and 4.3 we discuss how the bounds grow with rate $\eta T$ in the setting of Section 3 and with rate $\sqrt{\eta T}$ in the setting of Section 4 when the number of iterations $T \ll 1/\eta$.
>
>
> ## A.3 – How do your assumptions on the learning rate relate to practical choices?
> Jastrzębski et al. [*] provide empirical evidence that the generalization capabilities of SGD are determined by the ratio $\eta / k$. In particular, they show that as $\eta / k$ decreases, validation accuracy worsens. With this, our assumptions on the size of $n$ are unsurprising: if $\eta / k$ is made smaller we require more data to observe good generalization.
>
> Since it is often claimed that the discretization error needs to be small for SGLD to succeed it is usually assumed that $\eta$ scales with the target training accuracy [22, 31]. Thus, our assumption that $\eta$ scales with $1/n$ is consistent with existing theoretical results.
>
>
> ## A.4 – Is weight decay needed for the analysis?
> In its most general form (Section 4), our result assumes that the loss function is dissipative and smooth and does not require the use of weight decay regularization. As a special case, to obtain sharper bounds, we consider Lipschitz smooth loss functions with weight decay (Section 3).
>
> Without weight decay, no Lipschitz function can satisfy the dissipativity assumption needed to facilitate contractions and so weight decay is needed in this special case. Theorem 3.1 supports all choices for $\lambda \in \mathbb{R}^+$ but the bound diverges as $\lambda \to 0$.
>
>
> ## A.5 – Why is the weight-decay in the form of $\ell_2$, rather than $\ell_2^2$ norm?
> This is a typo, the weight decay regularization we consider is indeed the $\ell_2^2$ norm. Thank you for pointing this out.
>
>
> ## References
>
> [*] Jastrzębski S, Kenton Z, Arpit D, Ballas N, Fischer A, Bengio Y, Storkey A. Three factors influencing minima in sgd. arXiv preprint arXiv:1711.04623. 2017 Nov 13.
>
> [**] Yi M, Wang R, Ma ZM. Characterization of Excess Risk for Locally Strongly Convex Population Risk. arXiv preprint arXiv:2012.02456. 2020 Dec 4.

---

> ### Author Response · Authors · 2021-09-09
> **Response Follow-up**
>
> Dear Reviewer,
>
> Just a quick follow up regarding our response to your comments. Specifically, If you have any other outstanding concerns, do let us know as we can provide additional clarification.
>
> Otherwise, if you have no other concerns, we would kindly ask that you consider updating your review in light of this.
>
> Best,
> Authors

---

### Comment · Area_Chair_rSM8 · 2021-08-22
**Please help with clarifying issues**

Dear authors,

I would like to ask for your clarification on two concerns regarding the generalization bounds established in the paper:

- It is unclear to me how significant are the $\sqrt{d/n}$-type generalization bounds proved in the paper: unless I am overlooking something, they seem to follow straightforwardly from standard uniform convergence / covering arguments (that imply bounds of the form $\sqrt{(d/n) \log(LDn)}$ for $L$-Lipschitz losses over a unit ball of radius $D$).  Granted, one has to bound the “effective” Lipschitz constant and diameter, but the dependence on these is logarithmic so very weak bounds suffice.

- The dimension-free bounds, on the other hand, appear to behave in a rather strange way w.r.t. $\beta$: the dependence is only through the ratio $d/\beta$ (as you stated in your response, so that by taking $d \propto \beta$ they become dimension-free); therefore, the bounds improve as $\beta$ becomes larger and larger, in which case SGLD becomes closer and closer to plain SGD.  That is, these results suggest that noiseless SGD generalizes well uniformly over time and independently of the dimension (this is already a very surprising result---even for convex SGD) and that the added noise in SGLD is not required for (and does not improve) generalization.

    However, I would have expected that stability, and therefore generalization, will actually improve as the noise increases, not the other way around, and indeed, this is the case for existing results for SGLD with bounds that increase as $\beta$ becomes larger.  Further, it is somewhat strange that the results hold with arbitrarily small noise, given that the tools used to obtain them rely strongly on distributional stability induced by the added noise.

    It is maybe natural to assume that there should be a bounded range of allowable $\beta$ and one cannot take it as arbitrarily large for the result to hold true; but if that is the case, it should be confirmed that $\beta \propto d$ is an admissible setting (and which indeed gives rise to dimension-free bounds).

I would appreciate your prompt assistance with clearing out the above concerns.  Thanks!

AC

---

> ### Author Response · Authors · 2021-08-29
> **Response to Area Chair rSM8**
>
> We thank the AC for their questions, which we address below.
>
> ### $\sqrt{d/n}$ dependence in Theorem 3.1
>
> A generalization error bound of the order $\sqrt{d/n}$ can be recovered by bounding the Rademacher complexity for *linear* models constrained over a Euclidean ball with Lipschitz *loss functions*, for example, either directly (applying Cauchy-Schwarz) or through covering number arguments (and using chaining to remove log terms). In general, Lipschitz predictors (and not just loss functions that are Lipschitz in the parameters) are known to lead to polynomial metric entropy spaces (as opposed to logarithmic spaces), so yielding a larger dependence on the dimension.
>
> ### Theorem 3.1 correction
>
> In the response to reviewer v2Zx, we have outlined a sketch of a modification of the proof of Theorem 3.1 so that the bound can be made dimension-free, matching the dimension-free nature of Theorem 4.1 when $\beta \propto d$. While working on the details to include the final result in our paper, however, we have now noticed that there seems to be a problem with our proposed strategy to modify the proof and that the dimension-dependence nature of that result appears to be fundamentally linked to the approach that Eberle [8] uses to attain the $W_1$ contraction, which is what allowed us to establish $1/\sqrt{n}$ rates (one of the main objective of our work is to establish time-independent bounds that go to zero as n increases). We are exploring this more in depth, and will include an extensive comment on this in the camera-ready version of our work. We are now going to follow up with reviewer v2Zx on this, so they are also aware.
>
> Nevertheless, we believe the original result as presented in our submission is already of interest despite its dependence on $d$ (c.f. the previous point). We have also included this result in the paper for two additional reasons. Firstly, because it helps us establish the methodology used in a way that is simpler than the more general case. Secondly, we obtain better rates w.r.t. n; this is at the expense of bad dimensional dependence, at least in this case.
>
> ### Dependence w.r.t. $\beta$
>
> Note that in Theorem 4.1, the upper bound we have developed for the case of dissipative losses blows up in the limit for $\beta$ going to zero and in the limit for $\beta$ going to infinity.
>
> The fact that our upper bound explodes as $\beta$ grows to infinity is in line with the intuition of the AC that adding noise should improve generalization. The AC is correct in noting that other works have provided upper bounds that improve as $\beta$ goes to zero, but these bounds also increase indefinitely with time, as opposed to our time-independent bounds. In our work, we upper bound the quantity $\sup_{t \in \mathbb{N}} \mathbb{E} gen(x_t)$ and the bound we obtain is smallest when $\beta$ is not too large or too small, forming a sort of u-shape in the parameter $\beta$.
>
> The above-mentioned dependence on $\beta$ in our work can be seen by looking at the definition of the constants $C_5$ and $C_6$ in the proof of Proposition 4.2 in Appendix E.1. In particular, $C_11$ and $C_11/C_10$ explode as $\beta$ goes to zero while the contraction rate $\dot{c}$ goes to zero as $\beta$ goes to infinity (see equations (12), (13) and (14)). We plan to make this dependence more explicit in the camera-ready version of our paper, presenting a discussion on the noise term and how our bound is qualitatively different from previously-developed bounds for SGLD in this respect.
>
> ### Other remarks
>
> We would like to take the opportunity to stress that, within the literature that have looked at generalization bounds for SGLD, the novelty of our results stems from the following two main points. Firstly, our bounds support the dissipative smooth setting that has become increasingly popular in this field. Secondly, our bounds are uniform over all time-steps, which comes as a result of our use of the concentration phenomenon of SDEs.

---

### Decision · Program_Chairs · 2021-09-27

**Decision:**

Accept (Poster)

**Comment:**

The paper has been received with mixed impressions. I mostly agree with the bottom-line of the reviews: the paper does introduce novel results and techniques to the line of work on generalization of SGLD, and there should be sufficient interest at NeurIPS for such a contribution; yet on the other hand, it is unclear how significant are the actual bounds derived, and the paper does not provide ample discussion/comparison around this.

As some of the reviews alluded to, the significance of the dimension-dependant bounds is questionable (under the assumptions the paper makes, $\sqrt{d/n}$-type bounds can be obtained, independently of the algorithm, from generic uniform-convergence arguments; the authors response to this point is unconvincing).  While the dimension-independent bounds (Theorem 4.1) seem more compelling, for truly appreciating them I would have liked to see a more nuanced discussion of their relation to existing bounds, as well as a more careful analysis of their precise dependence on the problem parameters.

All considered, my decision is to accept the paper - but I urge the authors to carefully address the primary concerns raised in the discussion for the final version and carefully clarify their contribution and its relation to existing generalization bounds.